# Multi-modal cryo-EM reveals trimers of protein A10 to form the palisade layer in poxvirus cores

Julia Datler ®[1,2], Jesse M. Hansen[1,2], Andreas Thader[1], Alois Schlögl ®[1],
Lukas W. Bauer[1], Victor-Valentin Hodirnau ®[1] & Florian K. M. Schur ®[1] ✉

Poxviruses are among the largest double-stranded DNA viruses, with members such as variola virus, monkeypox virus and the vaccination strain vaccinia virus (VACV). Knowledge about the structural proteins that form the viral core has remained sparse. While major core proteins have been annotated via indirect experimental evidence, their structures have remained elusive and they could not be assigned to individual core features. Hence, which proteins constitute which layers of the core, such as the palisade layer and the inner core wall, has remained enigmatic. Here we show, using a multi-modal cryo-electron microscopy (cryo-EM) approach in combination with AlphaFold molecular modeling, that trimers formed by the cleavage product of VACV protein A10 are the key component of the palisade layer. This allows us to place previously obtained descriptions of protein interactions within the core wall into perspective and to provide a detailed model of poxvirus core architecture. Importantly, we show that interactions within A10 trimers are likely generalizable over members of orthopox- and parapoxviruses.

Poxviruses are large, pleomorphic, double-stranded DNA viruses that infect a wide range of hosts, from vertebrates, including humans, to arthropods[1]. Among the members of the Poxviridae family are variola virus, the causative agent of smallpox, and VACV, the prototypical and most extensively studied poxvirus. VACV was also used as an attenuated vaccination strain to eradicate smallpox in the late 1970s[2]. Recently, the re-emergence of monkeypox virus, which has caused localized outbreaks of mpox around the globe, has re-emphasized the importance of a better understanding of the intricate poxvirus lifecycle.

Poxvirus replication occurs within viral factories that are exclusively located within the cytoplasm of a host cell and gives rise to immature viruses (IVs), which eventually transition into infectious intracellular mature virions (MVs) and extracellular enveloped mature virions (EVs)[3]. MVs are enveloped by a lipid bilayer and contain a dumbbell-shaped core that encapsulates the viral DNA genome, and lateral bodies (LB), which laterally attach to the exterior of the core

wall and contain viral proteins for modulating host immunity and the oxidative response[1,4,5] (Fig. 1a). The transition from IVs to MVs requires proteolytic cleavage of several core proteins, which in turn contribute to the formation and condensation of the viral core, with its characteristic shape and biochemical signature[6,7]. The core is one of the uniting factors in all the infectious poxvirus forms and fulfills one of the key roles in the virus lifecycle, that is the protected transfer of the viral genome and required accessory proteins to a newly infected cell. Because of this, substantial effort has been invested into detailed structural and biochemical characterization of the core. However, the structural determinants that underlie core morphogenesis have remained poorly understood, impeded by the molecular complexity of poxviruses and the apparent lack of sequence homology of the suggested structural protein candidates to other species.

Room-temperature and cryo-electron microscopy (cryo-EM) analysis of VACV has revealed the presence of a regular, but discontinuous,

[1]Institute of Science and Technology Austria (ISTA), Klosterneuburg, Austria. [2]These authors contributed equally: Julia Datler, Jesse M. Hansen.
✉e-mail: florian.schur@ist.ac.at

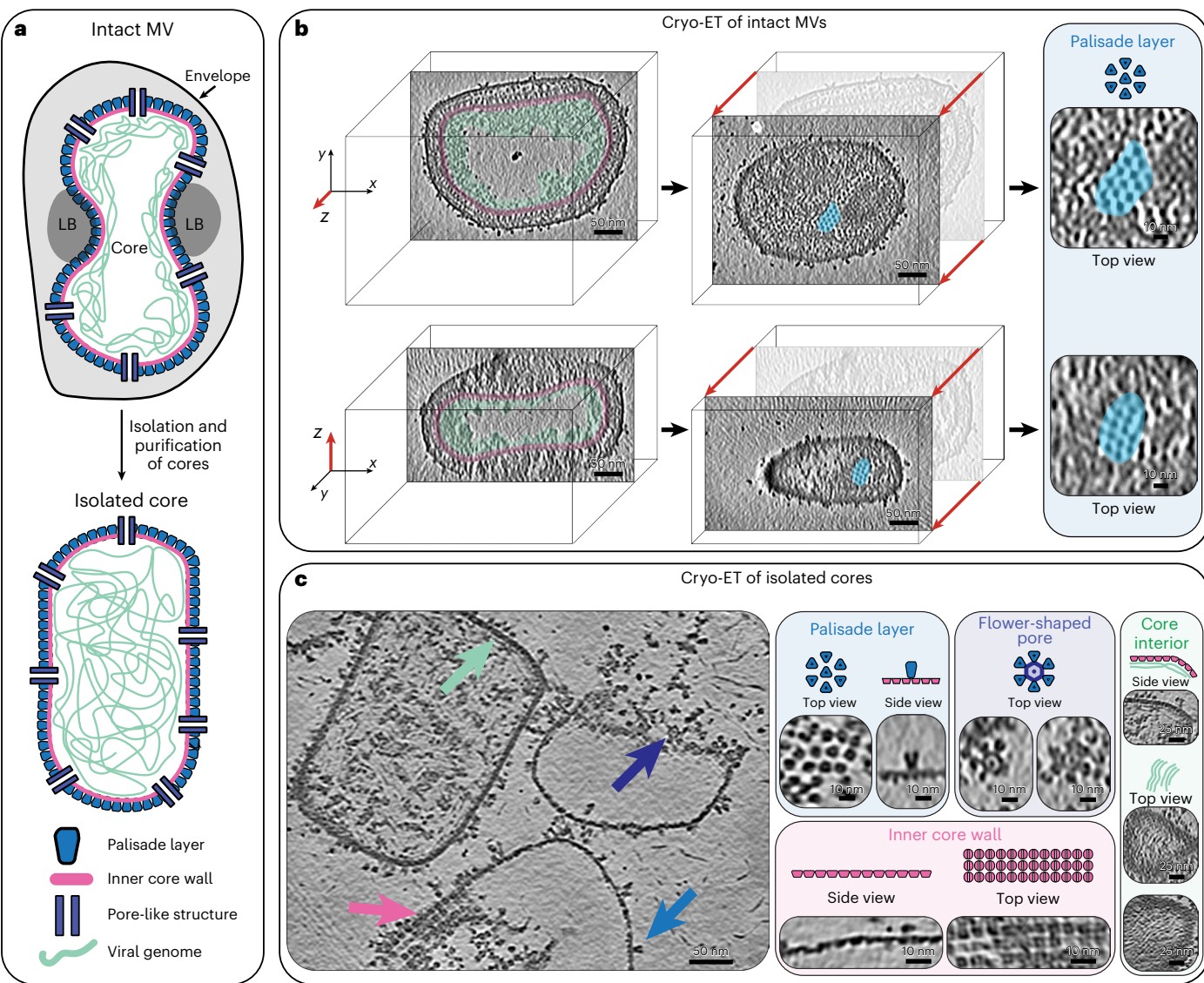

**Fig. 1 | Cryo-ET of VACV mature virions and isolated cores. a**, Schematic of intact MV (top) and an isolated MV core (bottom), showing previously described structural entities, such as the palisade layer, inner core wall, pore-like structure and viral genome. Lateral bodies (LB) bind to the concave-shaped core in intact viruses, but are often lost during core isolation. This color coding for the palisade layer (blue), the inner core wall (pink), the pore-like structure (purple) and the genome (green) is kept consistent throughout subsequent figures. **b**, Computational slices (1.1 nm thickness) through a missing wedge-corrected tomogram (using IsoNet) of an intact MV particle (this slice is representative of 15 tomograms), clearly showing core morphology and structural features such as the hexameric arrangement of the palisade layer (highlighted in blue), the inner core wall (pink) and the condensed genome (green). The virus is shown from two viewing directions, looking at the *xy* and *xz* planes (see axes on left) on top and bottom, respectively. The center panel shows the palisade layer in grazing slices, with a magnified view of this region shown on the right. **c**, Computational slice (1.1 nm thickness) through an IsoNet-corrected tomogram of isolated MV cores (left) (this slice is representative of four tomograms). Structural features are clearly observable in tomograms of isolated cores (annotated with arrows, same color scheme as **a**) and are shown on the right.

palisade layer formed of spike-like assemblies on the outside of a continuous viral inner core wall[8–10]. Improved imaging modalities using cryo-electron tomography (cryo-ET) have further suggested that the spikes of the palisade layer form a pseudo-hexameric lattice[11,12]. In addition, several studies have suggested the presence of pore-like structures with unknown function spanning the core wall[10–12].

Beyond these morphological descriptions of the core architecture, identities of proteins forming the core wall and palisade layer have been derived from studies using (1) immunogold labeling of intact VACV MV and purified cores[9,10,13–16]; (2) biochemical and proteomic studies via extraction and partitioning of MV components[5,17–19]; and (3) genetic experiments via recombinant viruses and inducible mutants of selected protein candidates[16,20–23]. Specifically, protocols to isolate intact cores

from native virus particles (Fig. 1a), using non-ionic detergent under reducing conditions, allowed for a more straightforward biochemical and ultrastructural description[5,8,10]. Together, these studies have offered an inventory of the proteins that presumably form the core wall and palisade layer, namely the cleavage products of precursor protein A10 (p4a in the UniProt database), precursor protein A3 (p4b in the UniProt database), A4 (p39 in the UniProt database) and L4 (VP8 in the UniProt database) (Extended Data Fig. 1a,b). For consistency with the current use of protein nomenclature in the poxvirus field, we will exclusively refer to the core proteins as A10, A3, A4 and L4. The minor cleavage product of A10 will be referred to as 23K.

In immunogold labeling experiments, A10 and A4 have been detected at the outer surface of the core wall[9,10,14–16]. Proteins A3 and A4

have been identified as components of the palisade layer[14,24]. Peculiarly, protein A3 has also been suggested to be located at the inner part of the core wall[10]. Furthermore, immunolabelling of cryosections[9] and broken viral cores[10] has found that L4 is located at the inner core wall, in line with its role as a major DNA-binding protein[22,25]. However, given these also partially contradictory results, no direct structural proof could be obtained to unambiguously assign any of these protein candidates to specific structural core features, that is the palisade layer, the inner core wall or protein densities in the core interior. Without this knowledge, key steps of the poxvirus lifecycle remain enigmatic, limiting possibilities for extending the potential tools for pharmacological interference during poxvirus infection.

Here we used a combination of cryo-ET, subtomogram averaging (STA) and single-particle cryo-electron microscopy (cryo-SPA) to study complete VACV MVs and isolated VACV virus cores. Our results show that the palisade layer and inner core wall adopt two different local symmetries, and we identify several distinct structural entities in the core. Importantly, our integrated use of cryo-ET and SPA, combined with AlphaFold[26], identifies that trimers of A10 form the palisade layer. Together, these results allow us to extend the structural atlas of poxvirus cores and present a substantially refined model of poxvirus core assembly.

## Results

### Cryo-ET of VACV mature virus

No experimentally derived structures of the major structural core protein candidates are available. Therefore, we used AlphaFold[26] to computationally predict models of the main core protein candidates, A10, 23K, A3, A4, and L4 (Extended Data Fig. 1c), to facilitate the interpretation of cryo-EM densities obtained in our downstream workflow. With the exception of A4, which was predicted to have a largely disordered fold, the other proteins adopted folds with good prediction certainty and defined secondary structure. The 23K protein formed an extended triple helix conformation, and L4 had a globular architecture. The topology of A10 formed an intricate fold, with the amino terminus being positioned centrally between two domains (Extended Data Fig. 2a,b). A3 displayed a compact shape, with its N-terminal and carboxy-terminal end located both on the same side of the protein structure (Extended Data Fig. 2c,d).

Foldseek[27] and Dali[28] analysis revealed no similarities between the protein fold of A10 and those of other cellular or viral proteins in the Protein Data Bank (PDB) or AlphaFold databases. Interestingly, this search showed that the highest similarity for A3 was to deubiquitinating proteins, as recently suggested[29]. For the other proteins, Foldseek and Dali hits had low probability or TM scores, further highlighting the structural dissimilarity between these proteins and cellular proteins or other viral proteins outside of the poxvirus family. To gain a better understanding which regions of each protein may be conserved oligomerization interfaces, we also performed ConSurf analysis[30] on these five major structural proteins (Extended Data Fig. 3).

In our first attempt to structurally annotate the proteins forming the individual features of the viral core, we performed cryo-ET on intact VACV MV virions purified from infected HeLa cells (Table 1). Despite the relatively large virus dimensions (approximately 360 nm × 250 nm × 220 nm), our reconstructed tomograms allowed us to visualize fine details of virus particles, with their characteristic brick-shaped overall morphology and dumbbell-shaped core (Fig. 1b and Supplementary Video 1). In line with previous cryo-EM studies of MVs[11,12], the exterior of the core surface was coated with spikes of the palisade layer. The core lumen was predominantly empty, except for the condensed viral genome underlying the inner core wall. To obtain high-resolution structures of the individual layers of the core, we performed subtomogram averaging (STA) (Extended Data Fig. 4a). This revealed that the palisade layer was composed of a hexamer of trimers (Extended Data Fig. 4b), as suggested previously[12]. In our STA structure, we did not

observe clearly ordered densities for the inner core wall, suggesting that it does not have the same arrangement as the palisade layer. At this point, our structures obtained from intact virus particles had insufficient resolution (approximately 13 Å) (Extended Data Fig. 4c) to identify the proteins forming the trimers or the inner core wall. However, our alignment protocol still allowed us to clearly visualize the overall arrangement of the palisade layer into a large-scale pseudo-hexagonal lattice (Extended Data Fig. 4d). The lattice displayed large areas of continuous organization interspersed with gaps and cracks that broke the lattice into locally symmetric patches. We did not observe any obvious pentamer formation that could allow the complete closure of a hexagonal lattice. This is reminiscent of the incomplete hexagonal lattice observed, for example, with Gag proteins in immature retroviruses[31], and is also reminiscent of the poxvirus D13 pseudo-hexagonal matrix reconstituted in vitro[32,33] and visualized in vivo[34]. Using the initial mesh defined on the surface of the core wall to extract subtomograms, and the measured size of a trimer within a hexamer-of-trimers unit, we calculated an average number of approximately 2,280 trimers (s.d. = ±309, $n$ = 15 virions) to constitute the palisade layer, not considering the presence of gaps and cracks.

### Cryo-ET of isolated MV cores reveals their complexity

To improve the resolution of core structural features, we decided to reduce the complexity of our experimental system and therefore isolated VACV cores by optimizing established protocols using the detergent NP-40 and dithiothreitol (DTT)[8,35,36]. Dubochet and colleagues have shown that trimers can be released from isolated cores and visualized as isolated particles upon vitrification[8]. In our vitrified sample, isolated cores sometimes partially collapsed and appeared seemingly empty, as was reported previously[8]. However, they usually retained a regular barrel shape, and did not display the concavities observed within MVs (Fig. 1c and Supplementary Video 2). The genome filled the entirety of the viral core (Fig. 1c, left core), rather than being restricted to just the region underneath the inner core wall, as was seen in intact MVs (Fig. 1b). This indicates that core isolation potentially leads to genome decondensation.

Strikingly, in partially collapsed cores, several structural features were already visible in the individual tomograms. Given the three-dimensional (3D) nature of tomograms, we could assign these features to the individual layers of the core (Fig. 1c). In particular, we could visualize trimers in the palisade layer, which were seemingly still organized into a pseudo-hexagonal lattice. More importantly, we could, for the first time to our knowledge, observe the structural arrangement of the inner core wall (Fig. 1c, inner core wall panel; see also Supplementary Video 2). Each unit of the inner core wall adopted a square-like shape of approximately 7.4 nm × 7.4 nm (as measured directly from tomogram slices containing the inner-core-wall units), displaying at least twofold symmetry. This was consistent with our interpretation, based on intact viruses, that the inner core wall does not follow the organization of the palisade layer. We found a third main architectural feature, flower-shaped structures (diameter of approximately 29 nm) consisting of 'petals' and a central ring with hexameric symmetry and an inner diameter of approximately 11 nm. We consistently observed a strong density positioned within its center. These assemblies may be the core wall pores reported in previous lower-resolution cryo-ET data[11,12] and negative-stain EM images of isolated cores[10]. Directly below the inner core wall, linear densities could often be observed, organized into a parallel, striated pattern (Fig. 1c). The dimensions of these densities suggest that they are DNA.

### SPA reveals a diversity of structures in isolated cores

Given the high quality of the isolated cores and clear visibility of structural features, we reasoned that single-particle analysis (SPA) cryo-EM would allow us to further improve the resolution of our structures. We therefore acquired two SPA datasets (Fig. 2a,b). One contained

**Table 1 | Data acquisition and processing statistics for cryo-ET and SPA**

| | STA of A10 trimer from intact VACV virus (EMDB-17411), (EMDB-17413) | Cryo-ET VACV isolated cores (EMDB-17414) | A10 Trimer (residues 1–599) SPA (EMDB-17410), (PDB 8P4K) | Flower-shaped pore SPA (EMDB-17412) | A3 inner core wall SPA (EMDB-18452) | SPA VACV soluble fraction of isolated cores |
|---|---|---|---|---|---|---|
| **Data collection and processing** | | | | | | |
| Magnification | ×64,000 | ×64,000 | ×81,000 | ×81,000 | ×81,000 | ×81,000 |
| Voltage (kV) | 300 | 300 | 300 | 300 | 300 | 300 |
| Electron exposure (e$^-$/Å$^2$) | 165 | 165 | 53.06 | 53.06 | 53.06 | 80.0 |
| Dose rate (eps) | 18.59 | 22.74 | 24.458 | 24.458 | 24.458 | 23.43 |
| Defocus range (µm) | −1.5 to −8.0 | −1.5 to −5.0 | −1.25 to −3.0 | −1.25 to −3.0 | −1.25 to −3.0 | −1.5 to − 2.2 |
| Pixel size (Å) | 1.381 | 1.381 | 1.06 | 1.06 | 1.06 | 1.06 |
| Acquisition scheme / tilt | −66/66°, 3° | −66/66°, 3° | 0° | 0° | 0° | 25° |
| Frame number | 10 | 10 | 34 | 34 | 34 | 54 |
| Symmetry imposed | $C_3$ | – | $C_3$ | $C_6$ | $C_1$ | – |
| Tomograms for STA/ micrographs for SPA | 15 | n/a | 9,264 | 9,264 | 9,264 | 11,621 |
| Initial particle images (no.) | – | – | 224,331 | 26,127 | 211,924 | – |
| Final particle images (no.) | 27,922 | – | 24,943 | 14,330 | 18,452 | – |
| Map resolution (Å) | 13.1 | – | 3.8 | 7.2 | 20.7 | – |
| FSC threshold | 0.143 | – | – | 0.143 | 0.143 | – |
| FSCref threshold | | – | 0.5 | – | – | – |
| Map resolution range (Å) | 13.1–596.2 | | 3-7 | 5–15 | 20.7–542.7 | – |
| **Refinement** | | | | | | |
| Model resolution (Å) | – | – | 3.5 | – | – | – |
| FSC threshold | – | – | 0.143 | – | – | – |
| Model resolution range (Å) | – | – | 3.8 to infinity | – | – | – |
| Map sharpening *B* factor (Å$^2$) | −2,100 | – | – | −424.39 | – | – |
| Model composition | | | | | | |
| Non-hydrogen atoms | – | – | 14,568 | – | – | – |
| Protein residues | – | – | 599 | – | – | – |
| *B* factors (Å$^2$) | | | | | | |
| Protein | – | – | 99.79 | – | – | – |
| R.m.s. deviations | | | | | | |
| Bond lengths (Å) | – | – | 0.020 | – | – | – |
| Bond angles (°) | – | – | 2.618 | – | – | – |
| Validation | | | | | | |
| MolProbity score | – | – | 0.79 | – | – | – |
| Clashscore | – | – | 1.00 | – | – | – |
| Poor rotamers (%) | – | – | 0.00 | – | – | – |
| Ramachandran plot | | | | | | |
| Favored (%) | – | – | 98.99 | – | – | – |
| Allowed (%) | – | – | 1.01 | – | – | – |
| Disallowed (%) | – | – | 0.00 | – | – | – |

isolated viral cores, and the second retained only individual components released from cores as the sample was prepared with an additional centrifugation purification step before vitrification. Two-dimensional (2D) classification of these datasets revealed the structural treasure chest of the VACV core (Fig. 2c), yielding classes of trimeric, tetrameric, pentameric, and hexameric assemblies. Beyond classes for soluble particles, released either from the core wall or the core interior, we obtained classes of the flower-shaped pores which were still retained in the core wall (Fig. 2a). In addition, we obtained classes for continuous segments of the core wall. Some classes contained only the inner core wall, resembling in appearance the twofold symmetric assembly observed in tomograms of isolated cores. Other classes contained the inner core wall plus densities of the trimers in the palisade layer. Notably, we could not obtain classes in which both the inner core wall

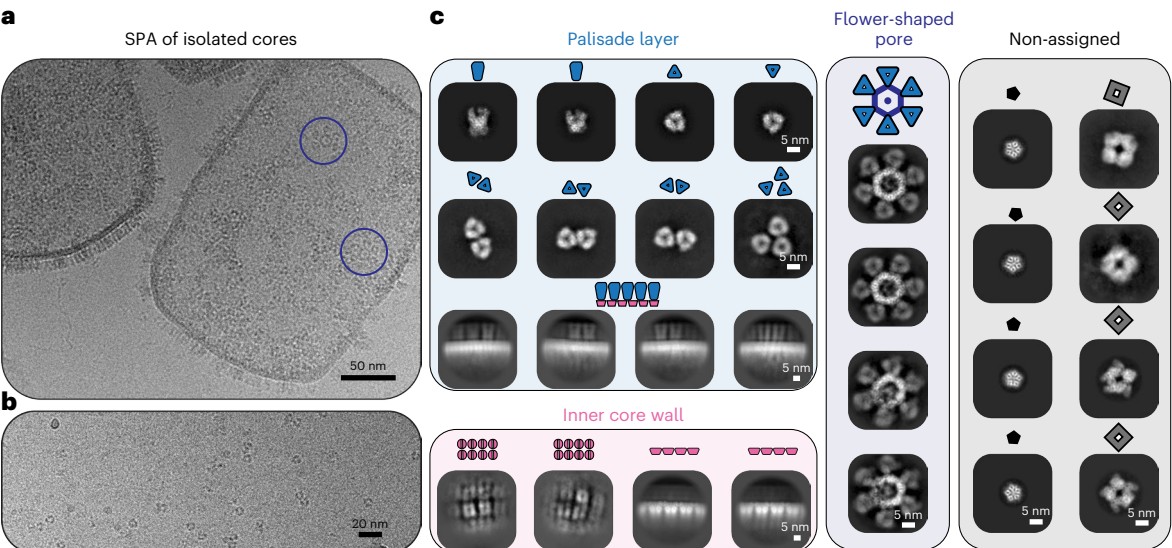

**Fig. 2 | The structural treasure chest of isolated VACV cores.**
**a**,**b**, Representative micrographs (of 9,264 in total) for the SPA acquisition, showing isolated cores (**a**) and soluble core proteins (**b**), such as trimers. The micrographs shown are from the same data acquisition. The flower-shaped pore, also observed in cryo-ET of isolated cores (Fig. 1c), is enclosed in purple circles. **c**, Gallery of 2D classes obtained from processing cryo-SPA datasets showing different multimeric assemblies observed as part of the core or as soluble components. Comparison of these 2D classes with our cryo-ET data allows a clear contextualization of their origin with respect to the palisade layer, the inner core wall or the flower-shaped pore. Classes for which no features could be observed in our cryo-ET data are labeled as non-assigned. The assembly and symmetry state of the classes is also annotated with small schematic depictions.

and the palisade layer were equally well resolved, again indicating that these two structural layers have independent organization with respect to each other. Given the wealth of structures within our sample, we aimed to determine the higher-resolution structures.

## Proteomics of isolated cores

First, to verify protein candidates of interest were retained following sample preparation, we performed mass spectrometry of the soluble protein fraction in the isolated core sample (Supplementary Table 1). Our proteomics data confirmed that our SPA sample was enriched for the major structural core proteins A10, A3, A4 and L4. In addition, we found many other proteins previously reported to be packaged into the core, including those involved in transcription and translation of the viral genome, as well as several host proteins.

## Trimers of A10 constitute the palisade layer

For structure determination, we first focused on the trimers in our SPA data, given their prominent structural appearance as part of the palisade layer. Using 3D refinement in RELION, we obtained a high-resolution reconstruction of the trimer at a global resolution of 4.2 Å, which we further improved using Phenix's density modification[37] to a global resolution of 3.8 Å (Fig. 3a and Extended Data Fig. 5). This map enabled visualization of structural detail such as bulky side chains and alpha-helical pitch (Fig. 3b), facilitating precise fitting of our computationally predicted models (Extended Data Fig. 1c) into our EM density. This unambiguously revealed that three copies of protein A10 (modeled residues 1–599) form the trimer (Fig. 3c and Supplementary Video 3), confirming a recent study using cross-linking mass-spectometry and modeling that suggested that A10 undergoes trimerization[38].

## The A10 trimer is stabilized through extensive interactions

Next, we used the PDBePISA server[39] to calculate properties and identify putative key contacts at the A10 trimer interface (Extended Data Fig. 6 and Supplementary Video 4). Given the resolution of approximately 4 Å, certain structural features, such as small and/or negatively charged side chains, are not clearly visible and hence limit interpretability.

Our analysis revealed that hydrophobic interactions dominate within the 2,104-Å$^2$ buried surface area per protomer pair (Extended Data Fig. 6a). There are also several conserved inter-chain salt bridges tethering central alpha helices together within the trimer (Extended Data Fig. 6b). Remarkably, each pair of protomers assembles a core heterodimeric three-stranded beta-sheet (residues 85–110, Fig. 3d and Supplementary Video 4), formed by two strands from one monomer and one strand from the neighboring monomer. A hydrogen bond network on the outward surface of the beta-sheet further reinforces this interaction, while the opposite side of the beta-sheet packs tightly with underlying hydrophobic side chains (Extended Data Fig. 6c).

A previous study suggested that VACV core proteins form disulfide bonds within MVs, helping to maintain the stability of released virus particles[40]. When analyzing our A10 model, we found two highly conserved cysteine residues (C31 and C569) in close proximity to each other (Extended Data Fig. 6d), which conceivably could clamp the N-terminal and C-terminal domains of A10 monomer together, stabilizing its conformation.

## The palisade layer forms weak interactions between trimers

The model of our trimer fits into our STA density map with high agreement, leaving no major area of density within the palisade layer unoccupied (Extended Data Fig. 7a). This strongly suggests that the A10 trimer is the main constituent of the palisade layer. However, our fit into the STA map also revealed that the lateral interactions across trimers within the palisade layer are not extensive, given the wide spacing between them. This further implies that the stabilization of the core does not solely rely on the A10 trimer but probably depends on the underlying inner core wall and additional interactions above the trimer.

Upon closer examination of our lattice maps of the palisade layer obtained through STA, we observed a variable orientation of individual trimers relative to each other (Extended Data Fig. 7b). This observation aligns with the existence of different multimeric trimer classes in the soluble fraction of the isolated core SPA dataset (Fig. 2c, Extended Data Fig. 7b), in which we observed interacting trimers exhibit substantial differences in their positioning relative to each other.

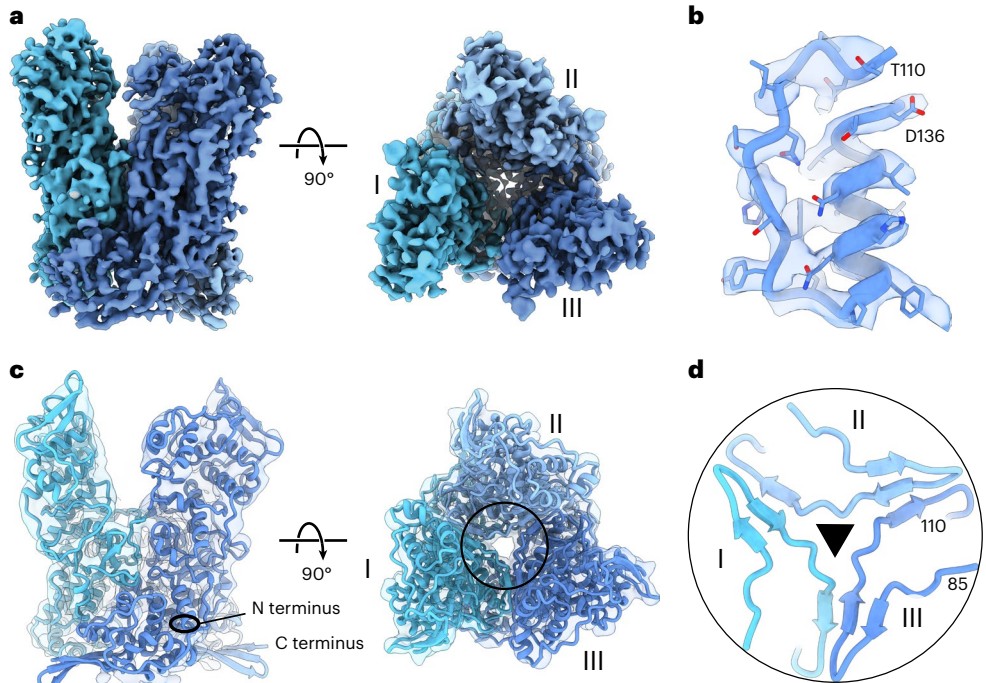

**Fig. 3 | Single-particle cryo-EM structure of the A10 trimer. a**, $C_3$-symmetric density-modified cryo-EM reconstruction of the A10 trimer at 3.8 Å resolution. Each subunit of the trimer (I to III) is depicted in a different shade of blue. **b**, The core region in the A10 trimer cryo-EM map is highlighted, where side chain density permits verification of the primary protein sequence. **c**, Refined model fit into cryo-EM density for the A10 trimer. The circle shows the zoomed-in region of interest in **d**. The transparent EM density has been low-pass filtered to 10 Å resolution to facilitate interpretation. The N terminus and C terminus of one of the A10 monomers are annotated. **d**, Central trimer contacts from **c**, showing residues 85–110, which engage in hetero-oligomer beta-sheet interactions with neighboring monomers. The residue numbers of the N terminus and C terminus of the displayed protein region are annotated.

## Positioning of core wall proteins with respect to the trimer

Given that A10 forms most, if not all, of the palisade layer, we wondered how other structural proteins in the core wall might interact with our A10 trimer structure, so we aimed to develop a spatial model of wall organization. We mapped previously reported mass spectrometry cross-linkages[19] onto our trimer model (Extended Data Fig. 8a). A4 preferentially interacts with the exterior side of the A10 trimer, in line with previous immunogold labeling experiments[9,10]. 23K and L4 interactions are mapped throughout the trimer model, although L4 interacts more often with centrally located residues. The major core protein A3 forms linkages exclusively with the interior side of the A10 trimer, suggesting that A3 could be a component of the inner core wall density we observe in our tomograms and SPA 2D class averages. Interestingly, the bottom of the A10 trimer facing the inner core wall is strongly positively charged (Extended Data Fig. 8b). Conversely, A3 has a negatively charged patch on one of its sides (Extended Data Fig. 8c). Indeed, when we used our SPA data to generate a low-resolution structure of the inner core wall density, an AlphaFold prediction of an A3 dimer fit into this density with good agreement (Extended Data Fig. 9). Although the low resolution and anisotropy of this map prevented determination of map handedness and unambiguous assignment of the orientation of A3 within the volume, our model, with A3's negatively charged patch facing towards A10 and its N terminus facing the core interior, satisfied the inner density most completely.

## The flower-shaped pore of the core wall

Next, we focused on the hexameric flower-shaped pore, which was positioned at the same height as the trimer within the core layer (Figs. 1c and 2c), but also further extended above as described[12]. Our cryo-EM reconstruction at 7.2 Å (Extended Data Fig. 10 and Supplementary Fig. 1) revealed the outer densities surrounding the center to

be the trimers of the palisade layer (Extended Data Fig. 10c). However, owing to map anisotropy, we were unable to unambiguously fit a model into the hexameric density at the core.

## A10 trimers are likely conserved in ortho- and parapoxviruses

The protein sequence of A10 is highly conserved among the *Orthopoxvirus* genus, including the key residues forming interactions in the trimer, with an average sequence identity of approximately 97% between VACV Western Reserve (WR), variola virus, monkeypox virus, rabbitpox virus, cowpox virus and ectromelia virus (Fig. 4a). In accordance with this high sequence conservation, AlphaFold predictions of these proteins as monomers or trimers are very similar (Fig. 4b).

More distantly related poxviruses show a substantially lower sequence identity in their A10 protein homologs. For instance, there is approximately 40% identity in the orf virus (a parapoxvirus) and approximately 22% in two members of the *Entomopoxvirus* genus, *Amsacta moorei* entomopoxvirus (AmEPV, over an alignment length of either 114 or 145 residues out of 1,149) and *Melanoplus sanguinipes* entomopoxvirus (MsEPV, over an alignment length of 293 residues out of 1,306) (Fig. 4a). AlphaFold predictions for the orf virus indicate a similar A10 trimer structure (Fig. 4b). AmEPV and MsEPV protein A10 homologs displayed a different overall fold. However, one region in AmEPV and MsEPV A10 homologs adopted a similar conformation compared with the base of A10 trimer in orthopoxviruses (Fig. 4c). This observation suggests that the region in A10 that is positioned towards the inner core wall might be a defining structural element among different distantly related poxvirus species.

Overall, these findings indicate that the interactions formed within the trimers constituting the palisade layer are similar among members of the *Orthopox* and *Parapoxvirus* genera, but potentially display interesting differences in comparison with entomopoxviruses.

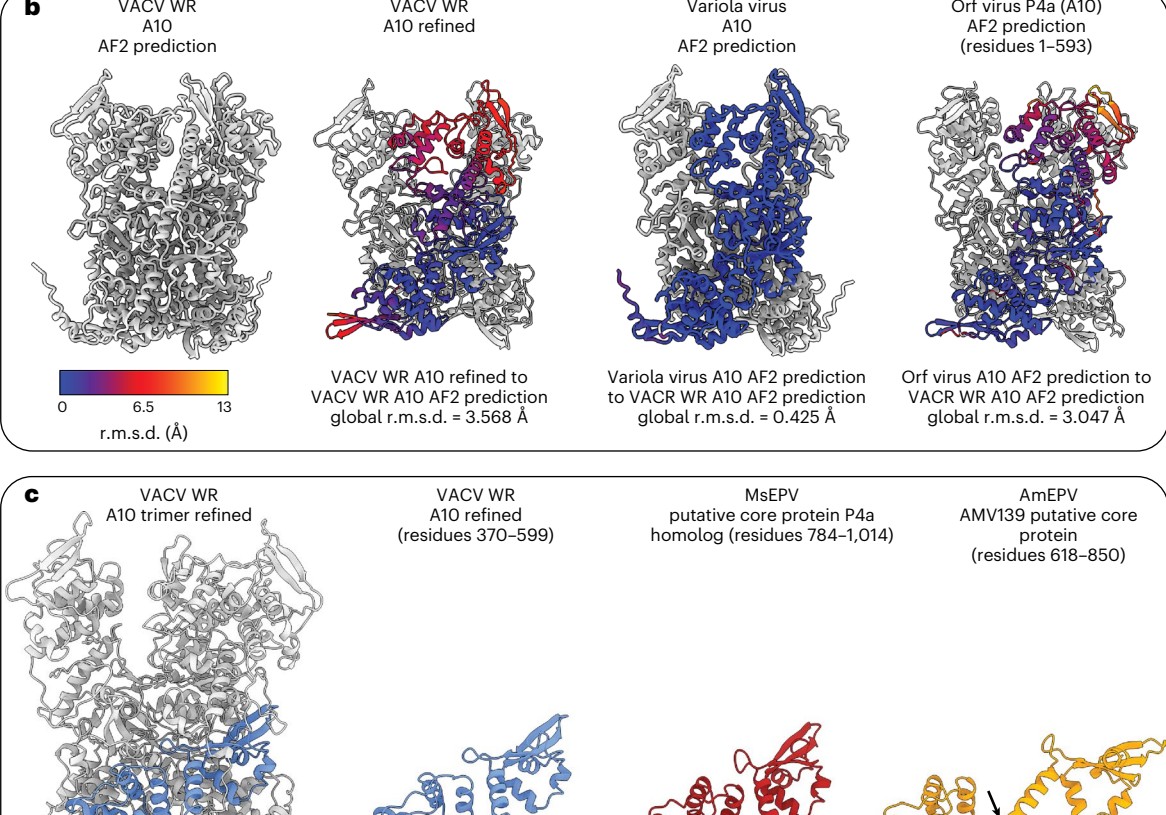

**a**

| Taxonomy | Genus | Protein | GenBank Protein ID | Alignment length (nt) | Sequence identity (%) |
|---|---|---|---|---|---|
| Variola virus | Orthopoxvirus | A10 | ABF23487.1 | 615 | 96.6 |
| Monkeypox virus | Orthopoxvirus | A10 | YP_010377118.1 | 614 | 96.6 |
| Rabbitpox virus | Orthopoxvirus | A10 | AAS49831.1 | 615 | 99.2 |
| Cowpox virus | Orthopoxvirus | A10 | ADZ29251.1 | 614 | 97.6 |
| Ectromelia virus | Orthopoxvirus | A10 | NP_671631.1 | 614 | 96.3 |
| Orf virus | Parapoxvirus | P4a (A10) | AY386264.1 | 511 | 39.5 |
| *Amsacta moorei* entomopoxvirus | Entomopoxvirus | AMV139 (putative core protein) | NP_064921.1 | 145/114 | 22.1 |
| *Melanoplus sanguinipes* entomopoxvirus | Entomopoxvirus | Putative core protein P4a homolog | AF063866.1 | 293 | 21.2 |

**b**

VACV WR A10 AF2 prediction | VACV WR A10 refined | Variola virus A10 AF2 prediction | Orf virus P4a (A10) AF2 prediction (residues 1–593)

r.m.s.d. (Å) 0 — 6.5 — 13

VACV WR A10 refined to VACV WR A10 AF2 prediction global r.m.s.d. = 3.568 Å

Variola virus A10 AF2 prediction to VACR WR A10 AF2 prediction global r.m.s.d. = 0.425 Å

Orf virus A10 AF2 prediction to VACR WR A10 AF2 prediction global r.m.s.d. = 3.047 Å

**c**

VACV WR A10 trimer refined | VACV WR A10 refined (residues 370–599) | MsEPV putative core protein P4a homolog (residues 784–1,014) | AmEPV AMV139 putative core protein (residues 618–850)

**Fig. 4 | Comparison of vaccinia virus Western Reserve A10 trimers to other members of the poxvirus family. a,** Sequence identity of A10 protein of different viruses of the poxvirus family, compared with VACV WR. **b,** Comparison of the initial VACV WR A10 AlphaFold (AF2) prediction with the refined VACV WR A10 structure (also shown in Figure 3), and AF2 predictions of variola virus A10 and the parapoxvirus orf virus P4a (A10) residues 1–593. This comparison shows the strong similarity between the protein folds between these virus species. It further shows that the biggest difference between the predicted and refined VACV WR A10 model is at the top of the trimer, facing the outside of the core.

The color code displays root mean square deviation (r.m.s.d.) variations, with lower values indicating that the structure is more similar. **c,** Comparison of the refined VACV A10 model with parts of the predicted putative core protein models of MsEPV and AmEPV. Analysis reveals that, despite an overall more different fold, MsEPV residues 784–1014 and AmEPV residues 618–850 adopt a highly similar fold compared with VACV A10 residues 370–599. In the AmEPV AF2 prediction, a slightly different angle in a connecting loop between two halves of the fold (annotated with an arrow) leads to a different orientation.

## Discussion

By using a combination of SPA to obtain a high-resolution structure of the trimer that constitutes the palisade layer, and cryo-ET to contextualize our observations of previously unidentified structural entities within VACV cores, we have unambiguously identified A10 as the key protein that forms the palisade layer of VACV cores. This finding now offers the possibility to place previously obtained descriptions of protein interactions and locations within the core wall into perspective and to provide a more detailed model of poxvirus core architecture (Fig. 5).

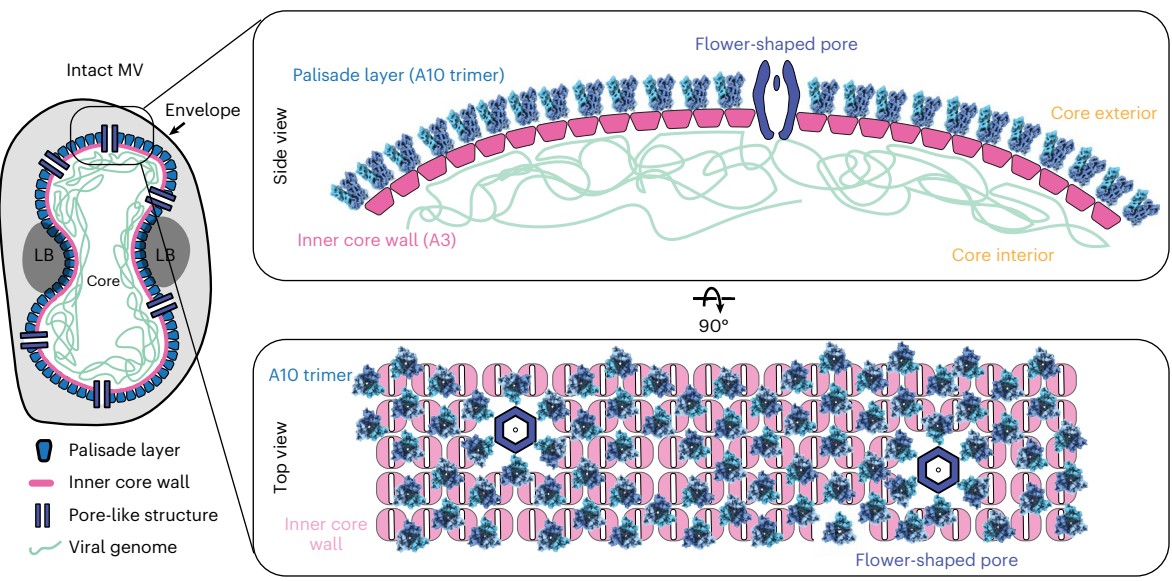

**Fig. 5 | Structural model of the VACV core wall.** Schematic summary of the updated model of the palisade layer and inner core wall. Protein A10 forms the palisade layer, which is positioned above an inner core wall with strikingly different symmetry. A4 is most likely decorating the outside of the palisade layer. The inner core wall is presumably formed by A3, with a potential role of L4 as a DNA-binding protein, tethering the viral genome to the core wall. The core is pierced by flower-shaped pores, which appear unevenly distributed on the surface of the core.

## The position of structural proteins within the core

Immunoprecipitation experiments showed that A10 and A4 form a stable complex even before proteolytic cleavage and MV formation[16]. Moreover, previous studies using immunolabeling speculated that the spike protein in the palisade layer was A4 (refs. 9,14,15). We show that A4 is not part of the trimers in the palisade layer, in line with its disordered nature predicted by AlphaFold (Extended Data Fig. 1c). Interaction sites between A4 and A10 are preferentially located on the core exterior-facing side of the trimer[19] (Extended Data Fig. 8a). This observation is consistent with a report claiming that A4 is on the exterior of the core, on the basis of data showing that A4 partially disappears after purification of the cores using the detergent NP-40 and DTT[10,15]. The extended conformation of A4 is reminiscent of minor coat proteins found in adenoviruses, termed cement proteins owing to their role in assembling and maintaining the virus shell[41,42]. Hence, A4 could have a role similar to that of these proteins, in which it provides additional stabilization to the palisade layer. A4 has previously been described as a matrix-like protein, which, for example, could establish a link between the core and surrounding membranes through binding to other viral membrane proteins[14].

Cross-linking mass spectrometry (XL–MS) data of VACV cores has also suggested that there is a direct interaction between protein A10 and A3 (ref. 19). On the basis of our model, these linkages are exclusively positioned on the bottom side of the trimer facing the core interior, suggesting that A3 is part of the inner core wall (Extended Data Fig. 8a). This is consistent with earlier experiments revealing that A3 is detectable only via immunogold labeling upon breakage of the core after hypertonic shock and protease treatment[10]. Accordingly, our low-resolution map of the inner core wall accommodates a dimer of A3. A small region of unoccupied density on the top of the inner core wall facing the palisade layer (Extended Data Fig. 9d, dashed rectangle) might be occupied by 23K, considering its placement in the proximity of the trimer based on XL–MS data. It is tempting to speculate on potential interaction interfaces between A10, 23K and A3 proteins on the basis of this observation, but more experimental work is needed to unambiguously define the positioning and the interactions between the individual core wall proteins.

L4 is a major DNA-binding protein[22,25] and has been reported to be located at the inside of the viral core[9,10]. In this regard, the predicted interactions between L4 and A10, based on XL–MS[19] data (Extended Data Fig. 8a) are not conclusive.

Considering these observations, we propose that the most likely arrangement of the major core proteins positions is A4 positioned on the outside of the viral core, A10 forming the palisade layer, as revealed in this study, and A3 building the inner core wall. Given that there is no unoccupied density in the palisade layer, 23K could be positioned either below the trimer or in the pore. Whether L4 might also be located in the pore or its central density remains to be determined.

## A10 as a shape-defining structural protein in the core wall

The finding that A10 forms trimers that constitute the palisade layer raises the question of their exact function. Our observation that the lateral interactions between trimers are not extensive and result in variable inter-trimer interactions within the lattice argues against a pure core-stabilizing role. The integrity of the viral core must therefore be achieved by another layer, such as the inner core wall, or aided by A4, which could link the trimers on the exterior of the viral core[14]. Instead, it is tempting to speculate that the trimers in the palisade layer determine the shape of the core, allowing it to form the highly complex dumbbell-shaped structure observed in MVs, with both convex and concave curvatures. In line with this hypothesis are conditional mutation studies that showed that, upon loss of A10, correct assembly of MVs is not possible[21]; instead, the inner core wall builds stacks or sheet-like architectures that can be labeled by antibodies to L4, F17 or E8 (ref. 20). This could suggest that A10 provides curvature-defining attributes to the viral core, while the inner core wall could act as a stabilizer through stronger lateral protein-protein interactions. This is also reflected in isolated cores, where trimers can be shed off the inner core wall while the core itself is held intact by the inner core wall.

## The poxvirus core pore

The flower-shaped pore identified in our cryo-EM and cryo-ET datasets most likely corresponds to the previously described pore-like structure[10–12]. Its function was postulated to be either directly involved in the mRNA extrusion into the cytoplasm of infected cells[10,11] or to be the

hexameric rings of the viral D5 primase and helicase, which is essential for vaccinia virus genome release[12]. Our cryo-EM density shows an additional density in the center of the flower lumen. Theoretically, the lumen could be large enough to accommodate DNA, but further experimental proof is needed to show whether the pore has a role in transporting nucleic acids through the core wall.

## Structural characterization of proteins from virus lysates

By working with isolated cores, which should contain all the components that the virus packs within them to infect new cells, we visualized several structural entities with varying symmetries. Given their abundance, they most likely have a relevant role in the viral lifecycle. This includes the pentamer, which was highly abundant in our data, and the tetrameric structures (Fig. 2c). Despite considerable efforts, we were not able to resolve structures, aside from the trimers, to high enough resolutions to unambiguously assign their identities. Future work could change the purification protocols for cores and improve sample vitrification protocols to reduce the observed preferred orientation to permit high-resolution structure determination of the other core components.

In vitro reconstitution of structural core proteins could also be pursued for the unidentified core classes. For example, this has been successfully used for structural studies of D13, which forms a hexameric lattice of the immature virus[32,33,43].

In a concurrent preprint[44] by Liu & Corroyer-Dulmot et al., the authors show that A10 forms trimers within the palisade layer. Classification of trimers on the surface of isolated cores further reveals an increased flexibility of the trimer, in line with our observations of variable inter-trimer interactions. Given the recent re-emergence of the monkeypox virus and the global epidemics that it has caused, our study and the concurrent work provide important fundamental and key insights into the poxvirus core architecture.

## Online content

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

## Methods

### Virus propagation and purification

VACV WR was received from A. Bergthaler (CeMM & Medical University of Vienna). VACV WR stock was trypsinized in a 1:1 dilution with 0.25% trypsin (Thermo Fisher Scientific, no. 25200056) for 30 min at 37 °C and 5% CO$_2$. HeLa cells (received from the Sixt lab, ISTA, which were originally purchased from ATCC) were seeded in DMEM GlutaMAX (Thermo Fisher Scientific, no. 31966047), supplemented with 10% (vol/vol) FBS (Thermo Fisher Scientific, no. 10270106) and 1% (vol/vol) penicillin–streptomycin (Thermo Fisher Scientific, no. 15070063) in T-175 flasks (Corning, no. CLS431080). Cells were washed once with PBS and infected with seed virus diluted in 2 ml infection medium (DMEM, 2.5% FBS, 1% penicillin–streptomycin) at a multiplicity of infection (MOI) of 1.5 and was incubated for 2 h at 37 °C and 5% CO$_2$ with manual shaking every 30 min. After 2 h, 33 ml of infection medium was added, and after 3 d, when a cytopathic effect was visible, cells were collected by scraping. The cell suspension was centrifuged at 1,200$g$ for 10 min at 4 °C, and the pellet was resuspended in 500 µl 10 mM Tris-HCl, pH 9 (Carl Roth, no. 9090.3) buffer. The cell pellet containing MVs was stored at −80 °C, and cells were lysed by three freeze and thaw cycles. Samples were centrifuged for 5 min at 300$g$ at 4 °C, and supernatant was collected. The cell pellet was resuspended in 500 µl 10 mM Tris-HCl buffer pH 9 and centrifuged again, and supernatant was again collected. Supernatants were pooled and applied on a 6 ml sucrose cushion of 36% sucrose (Sigma-Aldrich, no. 84100) in 10 mM Tris-HCl buffer, pH 9, in small tubes (Thermo Fisher Scientific, Thin-Walled WX, no. 03699) and centrifuged at 32,900$g$ at 4 °C for 80 min in an ultracentrifuge (Sorvall, WX100+, Rotor TH-641). The pellet was resuspended in 500 µl 1 mM Tris-HCl, pH 9, and applied on a sucrose gradient (40%, 36%, 32%, 28%, 24% in 1 mM Tris-HCl buffer pH 9) and centrifuged at 26,000$g$ at 4 °C for 50 min (Sorvall, WX100+, Rotor TH-641). The virus, visible as a milky band, was collected and sedimented in a fresh tube with 10 ml 1 mM Tris-HCl buffer pH 9, at 15,000$g$ for 30 min at 4 °C (Sorvall, WX100+, Rotor TH-641). The purified virus pellet was dissolved in 100 µl 1 mM Tris-HCl, aliquoted and stored at −80 °C until further use.

### VACV core purification

VACV core purification was optimized from previously described protocols[8,47]. As described above, the milky band in the sucrose gradient containing the virus was collected and sedimented in a fresh tube with 10 ml 1 mM Tris-HCl buffer pH 9 at 15,000$g$ for 30 min at 4 °C, and then dissolved in 500 µl core-stripping buffer containing 0.1% NP-40 (Thermo Fisher Scientific, no.85124), 50 mM DTT (Carl Roth, no. 69083), 50 mM Tris-HCl pH 9 and 2U DNAse (Promega, no. M6101). The virus was incubated for 10 min at room temperature and then centrifuged at 20,000$g$ for 30 min at 4 °C, through a 2-ml 24% sucrose cushion in 1 mM Tris-HCl, pH 9. Viral cores were collected in 500 µl 1 mM Tris-HCl buffer, pH 9, and sedimented at 15,000$g$ for 30 min at 4 °C. The final pellet was resuspended in 50 µl 1 mM Tris-HCl and frozen in aliquots at −80 °C until further use. For cryo-SPA samples of isolated VACV WR cores, four times more virus was used during purification, and 3 M KCl was added to a final concentration of 300 mM before being frozen at −80 °C.

### Purification of soluble fraction from isolated cores

Aliquots of isolated cores were thawed on ice, and 3 M KCl was added to a final concentration of 250 mM after all following dilution steps. Samples were mixed 1:3 with 0.25% trypsin and sonicated three times for 30 s each in a sonication bath (Elma, Elmasonic S40) at 4 °C, using the sweeping option. After 30 min of incubation at 37 °C, the samples were sonicated again three times for 30 s each. Samples were frozen at −80 °C and thawed at 37 °C, and sonicated again at 4 °C three times for 30 s each. Samples for cryo-SPA were centrifuged with 3,488$g$ for 45 min, and the supernatant was used for fixation and vitrification as described below. Samples for mass spectrometry were filtered through a 0.1-µm filter (Ultrafree MC-VV, Durapore PVDF 0.1 µm, Merck). The filter was prewetted with 1 M Tris-HCl and centrifuged for 3 min, at 12,000$g$ at 4 °C. Flow through was discarded, and the sample was added and centrifuged again. Flow through was stored at −80 °C until further use.

### Virus cryo-EM preparation and fixation

Whole VACV WR mature viruses and isolated cores were thawed on ice. For cryo-SPA samples of isolated VACV WR cores, 3 M KCl was added after thawing, resulting in a final concentration of 210 mM KCl after all of the following dilution steps. All samples, except the purified soluble fraction, were sonicated three times for 30 s each in a sonication bath at 4 °C with the sweeping option. Then, 0.25% trypsin was added in a 1:1 dilution and incubated for 30 min at 37 °C. All samples were fixed 1:1 with 4% PFA (Merck, no. P6148) (final concentration 2% PFA) in 1 mM Tris-HCl buffer, pH 9, and incubated for 30 min at room temperature and 30 min at 37 °C to inactivate the samples. Samples were frozen again at −80 °C and then sonicated three times for 30 s each in a sonication bath at 4 °C with the sweeping option before vitrification.

### Cryo-electron microscopy

BSA-Gold (10 nm, Aurion Immuno Gold Reagents, no. 410.011) in PBS was added to the cryo-ET samples in a dilution of 1:10. Samples for cryo-ET were deposited onto 300-mesh holey carbon grids (Quantifoil Micro Tools, R2/2 X-103-Cu300), and samples for cryo-SPA were deposited onto 200-mesh holey carbon grids (Quantifoil Micro Tools, R 2/2 X-103-Cu200), which were first glow-discharged for 2.5 min using an ELMO glow discharge unit (Cordouan Technologies). Then, 2.5 µl of the sample was added to both sides of the grid, which were vitrified using back-side blotting in a Leica GP2 plunger (Leica Microsystems). Blotting chamber conditions were 80% humidity and 4 °C. The grids were vitrified in liquid ethane (−185 °C) and then stored under liquid nitrogen conditions until imaging.

Singe-particle and tomography datasets were acquired under cryogenic conditions on an FEI Titan Krios G3i TEM microscope (Thermo Fisher Scientific) operating at 300 kV and equipped with a Bioquantum post-column energy filter and a Gatan K3 direct detector.

Cryo-electron tomography data were collected with the SerialEM software package version 3.8 (ref. 48). New gain reference images were collected before data acquisition. DigitalMicrograph 3.4.3, as integrated into the Gatan Microscopy Suite v3.3 (Gatan), was used for filter tuning, and SerialEM was used for microscope tuning. Tilt series were acquired with a filter slit width of 10 eV, using a dose-symmetric tilt scheme[49] ranging from −66° to 66° with a 3° increment. The nominal defocus range was set from −1.5 to −8 µm for the whole VACV WR mature virions and from −1.5 to −5 µm for the isolated viral cores. The nominal magnification was set to ×64,000, resulting in a pixel size of 1.381 Å. Tilt images were acquired in 5,760 × 4,092 pixel videos with 10 frames. The cumulative dose over the entire tilt series was 165 e$^-$/Å$^2$. For data acquisition settings, see Table 1.

The automated collection for the isolated viral core cryo-SPA dataset was set up using EPU version 2.13 (Thermo Fisher Scientific) in conjunction with AFIS. The soluble fraction purified from isolated cores was acquired using SerialEM version 4.0 (ref. 48), with an active beam tilt/astigmatism compensation. SPA micrographs were acquired in counting mode, with a filter slit width of 20 eV, and using a 4-shot-per-hole data collection. The nominal defocus was set from −1.25 to −3 µm for isolated cores and from −1.5 to −2.2 µm for the soluble fraction. The nominal magnification was set to ×81,000, resulting in a pixel size of 1.06 Å.

The isolated viral core dataset was acquired in 5,760 × 4,092 pixel videos with 34 frames, with a cumulative dose of 53.06 e$^-$/Å$^2$. The soluble fraction dataset was acquired with a tilted stage of 25 degrees in 5,760 × 4,092 pixel videos of 54 frames, with a cumulative dose of 80.20 e$^-$/Å$^2$. The decision to acquire tilted data was based on results from the

isolated core dataset, which revealed that several of the classes had a preferential orientation. Details for data acquisition can be found in Table 1.

## Image processing cryo-ET

The image processing workflow is schematically displayed in Extended Data Figure 4. Tomoman was used to sort and create stacks[50]. Defocus was estimated using CTFFIND 4.1.14 (ref. 51). IMOD 4.9.12 (ref. 52) was used for tilt series alignment and to generate ×8 binned tomograms with weighted back projection. The full tomograms were reconstructed in NovaCTF[53] with simultaneous 3D CTF correction with a slab thickness of 15 nm using the phase flip algorithm.

Bin8 tomograms were then filtered with IsoNet[54] to obtain better contrast and to fill missing information in the z dimension, in order to allow better visualization of the core (Supplementary Fig. 2). Definition of subtomogram averaging starting positions and all subsequent subtomogram averaging steps were performed in Dynamo version 1.1.333 (ref. 55). For starting positions for subtomogram averaging, we defined a mesh following the surface of viral cores in IsoNet-corrected bin8 tomograms. To generate a de novo reference, subtomograms (cubic size 464 Å³) were then extracted from IsoNet-corrected bin8 tomograms and subjected to five rounds of alignment with no symmetry applied. The first initial alignment reference was generated by averaging all particles, using their non-refined starting positions.

The obtained reference, which already displayed the hexamer-of-trimers arrangement, was then used to start a new subtomogram alignment in bin8, with subtomograms extracted at the initial mesh positions from weighted back projected tomograms (not IsoNet-corrected). $C_3$ symmetry was applied again only after the three-fold symmetry of the structure became clearly apparent upon initial bin8 iterations. Alignment was gradually refined from bin8 over bin4 (subtomogram cubic size 464 Å³) to bin2 (subtomogram cubic size 398 Å³) while advancing the low-pass filter and decreasing the Euler angle scanning step and range. After the first two alignments in bin8, overlapping particles were removed, using a distance cutoff of 6 pixels, and cc-threshold cleaned to remove subvolumes that did not align to the core surface. At the stage of bin2, the dataset was split into even/odd halfsets, and from this stage on, even/odd datasets were treated independently. Up to this point, the low-pass filter never extended beyond 25 Å. After the final bin2 iteration, the final halfset averages were multiplied with a Gaussian-filtered cylindrical mask, and the resolution was determined by mask-corrected Fourier-shell correlation[56]. The final map was sharpened with an empirically determined B factor of −2,100 Å² and filtered to its measured resolution[57].

## Image processing cryo-SPA

Videos from the dataset containing intact cores were motion-corrected with dose-weighting using the RELION 4.0-beta2 (ref. 58) implementation of MotionCorr2 with a patch size of 7 × 5. Motion-corrected micrographs were then imported into Cryosparc 4.0.0 (ref. 59) for subsequent processing. Processing details are summarized here, and full details are available in Extended Data Figure 5 (processing of the trimer) and Supplementary Figure 1 (processing of the flower-shaped pore). Initial CTF parameters were estimated using CryoSPARC patch CTF, and initial picks were obtained using a blob picker. Particles were extracted with a large box size (636 Å, bin4) early on to capture both large and small protein populations during 2D classification. Iterative 2D classification with varying mask sizes permitted sorting of particles into different protein species with distinct classes. Particles were then re-extracted, using a more appropriate box size: 340 Å for trimers, 545 Å for the flower-shaped pore and 636 Å for the side views of the core wall. In all cases, initial 3D volumes were generated using CryoSPARC ab initio without symmetry. Symmetry was then imposed during 3D auto-refinement.

Selected 2D classes containing particles for the flower-shaped pore were reconstructed using CryoSPARC non-uniform refinement,

and then were locally sharpened and filtered within CryoSPARC. Particles of the A10 trimer were exported to RELION for further processing: 2D classification, 3D refinement using a mask containing the full trimer density, Bayesian polishing, defocus refinement and focused 3D classification of A10 monomers. We manually balanced particle views of the trimer by removing over-represented top views during 2D classification, significantly improving map quality. Finally, the resulting map was density modified[37] using Phenix version 1.20-dev-4224 (ref. 60), with input of two half maps and the mask used during refinement. 3DFSC calculations were made using the Remote 3DFSC Processing Server (https://3dfsc.salk.edu). Local resolution for the A10 trimer map was calculated using RELION's implementation of ResMap[61], and cryoSPARC's local resolution estimate tool was used for the flower-shaped pore. Projections of the final flower-shaped pore were made using the V4 tool in EMAN/1.9 (ref. 62). The volume for the lower density of the core wall was generated in cryoSPARC from 3,795 side-view particles selected during 2D classification. A 2x binned initial model was generated ab initio without symmetry, and 3D non-uniform refinement without symmetry resulted in a final map at a global resolution of approximately 20.7 Å. We rigid-body fit the highest-ranking Alphafold2 model of the A3 dimer into the density using the UCSF Chimera (1.17.1) 'fit in map' tool. Options were set for real-time correlation using a simulated map at a resolution of 20 Å (estimated resolution from cryoSPARC) and optimization by correlation. Optimal orientation of the model was chosen on the basis of which option satisfied best the density as measured by UCSF Chimera's output correlation score of each fit.

The dataset collected at 25° stage tilt containing the soluble fraction was motion-corrected in RELION[58] identically to the dataset above. Data were processed separately from the dataset containing intact cores. CryoSPARC[59] patch CTF was used to estimate CTF to account for stage tilt. Particles were picked with a blob picker, as described above. Particles were extracted at bin2 with a box size of 340 Å and were subjected to iterative 2D classification with varying mask sizes and 250 classes.

## AlphaFold prediction of core protein candidates

Initial predicted structures of putative structural core proteins, either as a monomer or multimers, were generated using AlphaFold 2.3.2. (ref. 26) and Colabfold 1.5.2 (ref. 46). Five seeds were generated per model for orthopoxvirus proteins, and four seeds for the parapoxvirus and entomopoxvirus models, with one prediction per seed. All five models were relaxed using Amber relaxation and inspected manually. The highest-ranking model, as determined by pLDDT score, was selected for analysis. The following proteins were folded as monomers: A10 (UniProt P16715, residues 1–614), A3 (UniProt P06440, residues 62–643), 23K (UniProt P16715, residues 698–891), A4 (UniProt P29191) and L4 (UniProt P03295, residues 33–251) (Extended Data Fig. 1), orf virus p4A (GenBank ID AY386264.1), AmEPV AMV139 (GenBank ID NP_064921.1 putative core protein) and MsEPV (GenBank ID AF063866.1 putative core protein P4a homolog). Multimer predictions were made, with identical settings to those for monomer folds. The multimer prediction of the A10 trimer (UniProt. P16715, residues 1–614) and the predictions for the comparison of A10 trimer of variola virus (GenBank ID ABF23487.1, residues 1–615) and orf virus (GenBank ID AY386264 .1, residues 1–905) to VACV WR (UniProt P16715, residues 1–614) as well as the multimer prediction of the A3 dimer (UniProt P06440, residues 62–643) were done using AlphaFold version 2.3.2 and Colabfold[46].

## ConSurf analysis

ConSurf analysis[30] was performed using the web server https://con-surf.tau.ac.il/, with default parameters. Specifically, the amino acid sequence for each of the analyzed proteins was obtained via uploading the PDB file of the AlphaFold-predicted protein to the webserver. Similar sequences were then identified using PSI-PLAST (E-cutoff = 0.001).

For example, in the case of A10, this resulted in 31 sequences for comparison, including different members from the subfamily of Chordopoxvirinae (Orthopoxvirus, Parapoxvirus, Yatapoxvirus, Molluscipoxvirus among others), allowing a representative conservation analysis. A complete list of the included sequences is provided in Supplementary Table 2.

## Model building for the A10 trimer

Three copies of the highest-ranking A10 model (residues 1–599) were rigid-body fit into the density-modified trimer map shown in Figure 3a. The A10 trimer was refined and relaxed into the density-modified trimer with imposed non-crystallographic ($C_3$) symmetry through centroid relaxation in Rosetta/3.13 (20220812 build)[63–65]. Important parameters for refinement: resolution set to our global estimated value of 3.8 Å, and an elec_dens_fast density weight was set to a value of 3,5 as is suggested for maps at this resolution. The model was then imported into Coot (ver. 0.8.9.1 EL)[66] for per-residue manual inspection on a single chain, and real-space refinement was done if necessary. The chain was copied to other $C_3$ symmetry positions using UCSF Chimera[67]. Finally, the model was again refined using Rosetta, with NCS being applied. Model statistics were calculated using Molprobity integrated into Phenix[68] (Table 1). Pairwise r.m.s.d. calculations were done using the UCSF ChimeraX (version 1.5)[69] matchmaker tool. One final trimer model was rigid-body fit into the outer density of the flower-shaped pore and symmetrized sixfold using UCSF ChimeraX[69].

## Data visualization and figure preparation

Cryo-electron tomograms of whole VACV WR mature viruses and isolated cores, as well as the lattice maps, were visualized in IMOD[52], UCSF Chimera[67] and UCSF ChimeraX[69]. EM-densities were displayed in UCSF ChimeraX. Figures were prepared using Adobe Illustrator 2023. Videos were generated in UCSF ChimeraX and Adobe Premiere Pro 2023. The topology diagrams shown in Extended Data Figure 2 were generated using Pro-Origami[70].

## Proteomics

**Sample preparation.** To the soluble fraction from isolated cores (prepared as described above), 25 mM TCEP (Gold Biotechnology, no. 51805-45-9) and 4% SDS (Carl Roth, no. 8029,1) was added and boiled 10 min at 95 °C and was first cleaned up by SP3 using a commercial kit (PreOmics, 100 mg of beads per sample), then processed using the iST kit (PreOmics), according to the manufacturer's instructions. Tryptic digestion was stopped after 1 h, and samples were vacuum dried and then re-dissolved in the iST kit's LC LOAD buffer with 10 min sonication.

**LC–MS/MS analysis.** The sample was analyzed by LC–MS/MS on an Ultimate 3000 RSLC_Nano nano-HPLC (Thermo Fisher Scientific) coupled with a Q Exactive HF (Thermo Fisher Scientific), concentrated over an Acclaim PepMap C18 pre-column (5 µm particle size, 0.3 mm ID × 5 mm length, Thermo Fisher Scientific), then bound to an EasySpray C18 column (2 µm particle size, 75 µm ID × 50 cm length, Thermo Fisher Scientific) and eluted over the following 60 min gradient: solvent A, MS-grade $H_2O$ + 0.1% formic acid; solvent B, 80% acetonitrile in $H_2O$ + 0.08% formic acid; constant 300 nl min$^{-1}$ flow; B percentage: 5 min, 1%; 45 min, 31%; 65 min, 44%.

Mass spectra were acquired in positive mode with a data independent acquisition method: FWHM 8 s, MS1 parameters: centroid mode, 1 microscan, 120,000 resolution, AGC target $3 \times 10^6$, 50 ms maximum IT, 400 to 1,005 $m/z$; DIA scans: 24 MS2 scans per cycle, 57 windows of 11.0 $m/z$ width per cycle covering the range from 394.9319 to 1,022.21204 $m/z$ (−0.005 $m/z$ non-covered gap between adjacent windows), spectra acquired in Profile mode, with 1 microscan, at 30,000 resolution; AGC target $1 \times 10^6$, 60 ms maximum IT, NCE 27.

**Data analysis.** The raw file was searched in DIANN version 1.8.1 in library-free mode against *Homo sapiens* and vaccinia virus (strain Western Reserve) proteomes sourced from UniprotKB. Match-Between-Runs was turned off. Fixed cysteine modification was set to carbamidomethylation. Variable modifications were set to oxidation (M) and acetyl (protein amino terminus). Data were filtered at 1% FDR.

DIANN's output was re-processed using in-house R scripts, starting from the report table. Peptide-to-protein assignments were checked, then Protein Groups were assembled and quantified. The data are provided in Supplementary Data 1, and the filtered list exclusively showing the top hit proteins for VACV is shown in Supplementary Table 1.

## Reporting summary

Further information on research design is available in the Nature Portfolio Reporting Summary linked to this article.

## Data availability

The electron microscopy density maps of the A10 trimer and the hexameric flower-shaped pore, the subtomogram average of the palisade layer and representative tomograms for complete viruses as well as isolated cores have been deposited in the Electron Microscopy Data Bank under accession codes: EMD-17410, EMD-17411, EMD-17412, EMD-17413, EMD-17414 and EMD-18452.

The refined model of the A10 trimer has been deposited in the Protein Data Bank accession code: PDB 8P4K.

The UniProt codes of VACV core proteins used for structure prediction are: A10 (P16715), A3 (P06440), A4 (P29191) and L4 (P03295). GenBank Protein IDs of variola virus A10 (ABF23487.1), monkeypox virus A10 (YP_010377118.1), rabbitpox virus A10 (AAS49831.1), cowpox virus A10 (ADZ29251.1), ectromelia virus A10 (NP_671631.1), orf virus P4a (AY386264 .1), *Amsacta moorei* entomopoxvirus AMV139 (NP_064921.1) and *Melanoplus sanguinipes* entomopoxvirus putative core protein P4a homolog (AF063866.1) were used for protein sequence alignment. Source data for MS experiments are provided as Supplementary Information accompanying this manuscript.

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

## Acknowledgements

We thank A. Bergthaler (Research Center for Molecular Medicine of the Austrian Academy of Sciences) for providing VACV WR. We thank A. Nicholas and his team at the ISTA proteomics facility, and S. Elefante at the ISTA Scientific Computing facility for their support. We also thank F. Fäßler, D. Porley, T. Muthspiel and other members of the Schur group for support and helpful discussions. We also thank D. Castaño-Díez for support with Dynamo. We thank D. Farrell for his help optimizing the Rosetta protocol to refine the atomic model into the cryo-EM map with symmetry. F.K.M.S. acknowledges support from ISTA and EMBO. F.K.M.S. also received support from the Austrian Science Fund (FWF) grant P31445. This publication has been made possible in part by CZI grant DAF2021-234754 and grant https://doi.org/10.37921/812628ebpcwg from the Chan Zuckerberg Initiative DAF, an advised fund of Silicon Valley Community Foundation (funder https://doi.org/10.13039/100014989) awarded to F.K.M.S. This research was also supported by the Scientific Service Units (SSUs) of ISTA through resources provided by Scientific Computing (SciComp), the Life Science Facility (LSF), and the Electron Microscopy Facility (EMF). We also acknowledge the use of COSMIC[45] and Colabfold[46].

## Author contributions

Project administration: F.K.M.S.; supervision and funding acquisition: F.K.M.S.; conceptualization: J.D. and F.K.M.S.; methodology: J.D., J.M.H. and F.K.M.S.; investigation: J.D., J.M.H., A.T., V.-V.H. and A.S.; Software: A.S. and L.W.B.; validation, formal analysis, and visualization: J.D., J.M.H. and F.K.M.S.; data curation: J.D., J.M.H. and F.K.M.S.; writing—original draft: J.D., J.M.H. and F.K.M.S.; writing—review and editing: J.D., J.M.H., A.T., V.-V.H., A.S and F.K.M.S.

## Competing interests

The authors have no competing interests.

## Additional information

**Extended data** is available for this paper at https://doi.org/10.1038/s41594-023-01201-6.

**Correspondence and requests for materials** should be addressed to Florian K. M. Schur.

Putative major structural core proteins & their precursor proteins

**a**

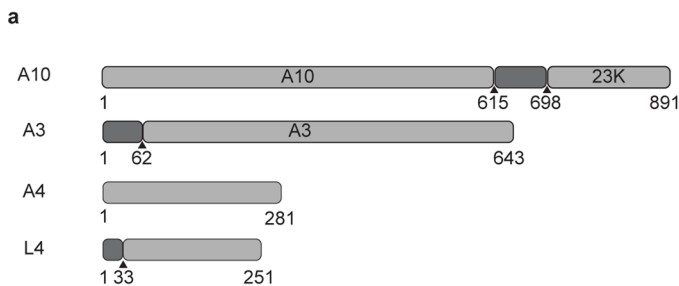

**b**

| Protein (UniProt name) | Residues | MW (kDa) | UniProt ID |
|---|---|---|---|
| A10 (4a) | 1 - 614 | 62 | P16715 |
| A3 (4b) | 62 - 643 | 60 | P06440 |
| 23K (cleavage product of A10 precursor) | 698 - 891 | 23 | P16715 |
| A4 (p39) | 1-281 | 31 | P29191 |
| L4 (VP8) | 33 -251 | 25 | P03295 |

**c**

AlphaFold-predicted models of major structural core proteins

**A10**
(1-614 Aa)

**A3**
(62-643 Aa)

**23K**
(698-891 Aa)

**A4**
(1-281 Aa)

**L4**
(33-251 Aa)

pLDDT score
0    50    70    90 100

**Extended Data Fig. 1 | Major structural proteins annotated to be in the VACV core wall. a)** Schematic representation of VACV WR proteins reported to constitute the core wall and palisade layer. Cleavage sites within precursor proteins are annotated by arrows and the resulting cleavage proteins are listed within the schematic representation. **b)** List of the structural core proteins discussed in this manuscript (in their cleaved form when applicable), with information on their residue length, molecular weight (kiloDalton) and UniProtID. **c)** AlphaFold-predicted models of A10, A3, 23K, A4 and L4. The modeled residues are indicated in brackets. The coloring of the model reflects the pLDDT confidence score (red = low confidence, blue = high confidence). The N- and C-termini of the proteins are annotated.

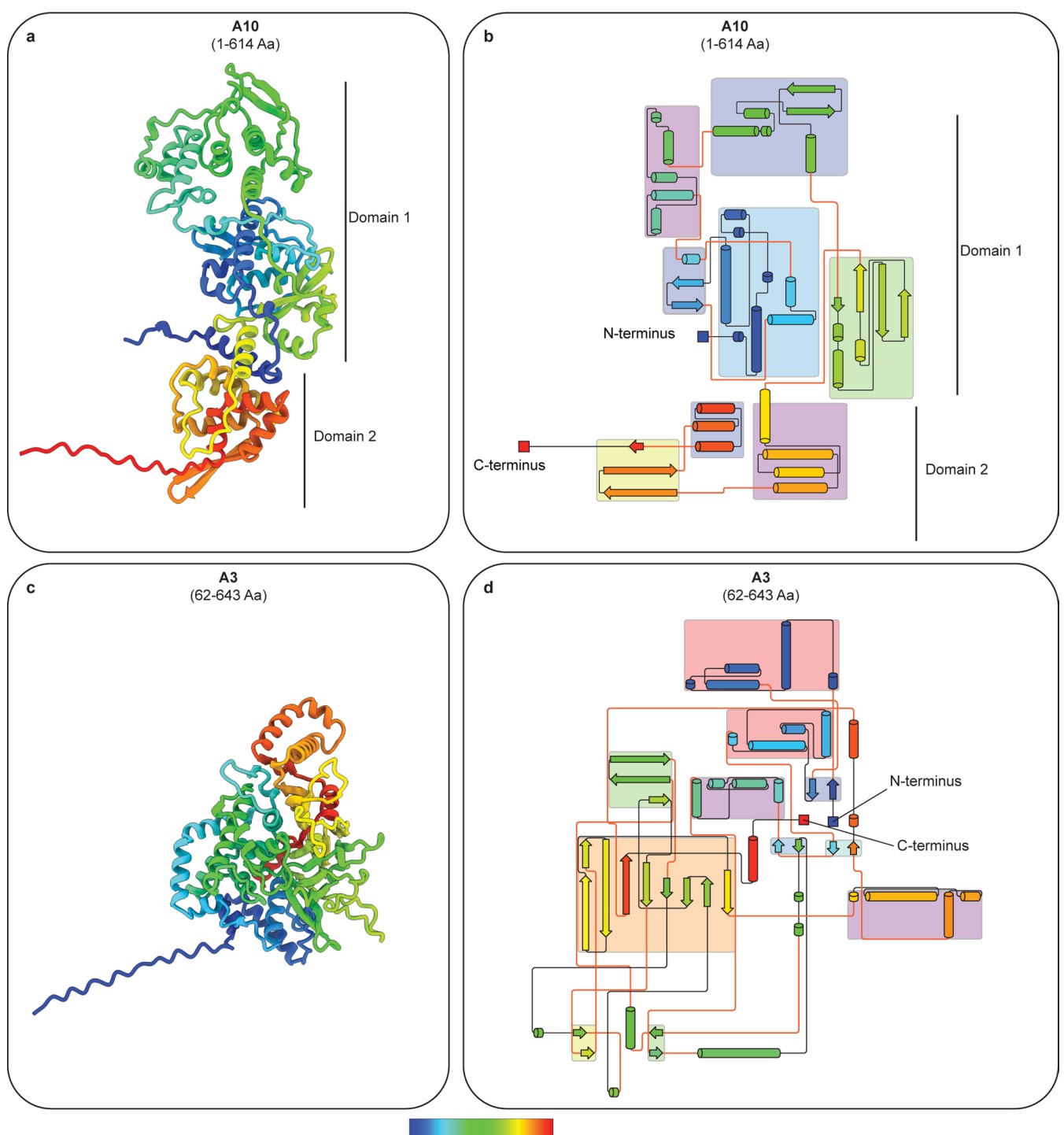

**Extended Data Fig. 2 | Protein topology of A10 and A3.** Illustration of protein topology and arrangement for A10 (panel a-b) and A3 (panel c-d). The coloring of the models and topology diagrams is according to residue positioning from the N-terminus (blue) to the C-terminus (red). **a)** Cartoon ribbon representation of the Alphafold-predicted model of A10, colored from N-terminus to C-terminus. The two sub-domains (SD1 and SD2) of the A10 fold are annotated. **b)** Protein topology diagram of A10, showing the positioning of protein regions, such as specific secondary structures with respect to each other. Beta-strands and alpha-helices are shown as arrows and cylinders, respectively. Their length is proportional to their number of residues. Colored boxes represent motifs which are positioned nearby relative to one another and therefore are likely to form a structural group. **c)** Cartoon ribbon representation of the Alphafold-predicted A3 colored from N-terminus to C-terminus. **d)** Protein topology diagram of A3, in the same depiction style as in (B). The topology diagrams shown in (b) and (d) were generated using Pro-Origami. The diagram in (b) was manually adapted to further improve visualization of the complex protein architecture. Orange line colors in (b) are used to highlight connections between structural groups. Orange lines in (d) are used to improve visualization but do not necessarily indicate connection between structural groups.

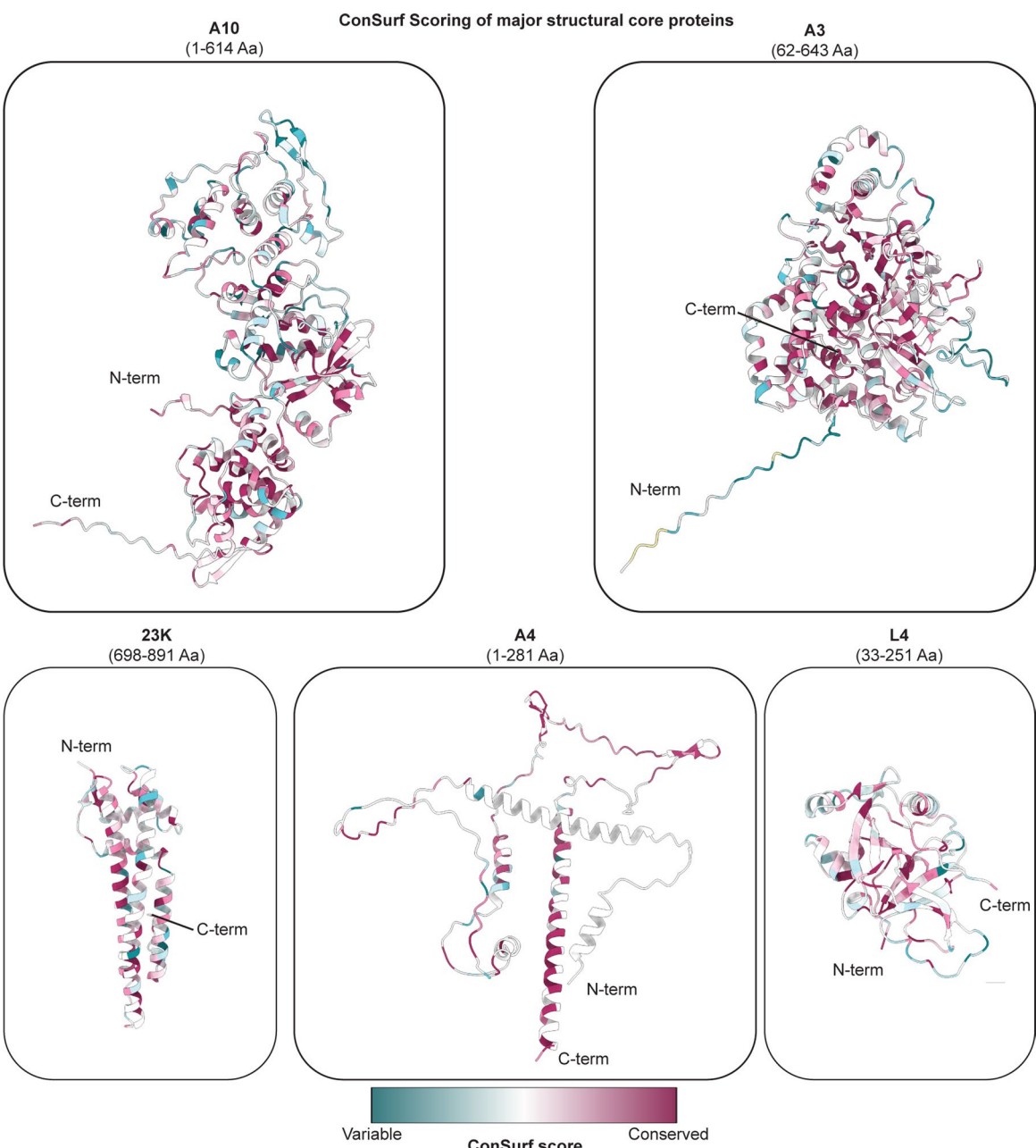

**Extended Data Fig. 3 | Sequence conservation analysis of structural core proteins.** AlphaFold-predicted models of A10, A3, 23K, A4 and L4, shown in the same orientation as in Extended Data Fig. 1c and colored according to their ConSurf score, highlighting the predicted conservation of individual residues. The modeled residues are indicated in brackets.

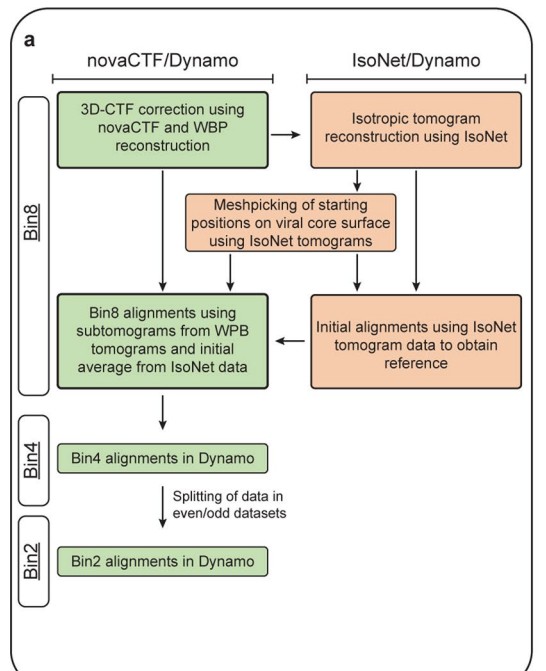

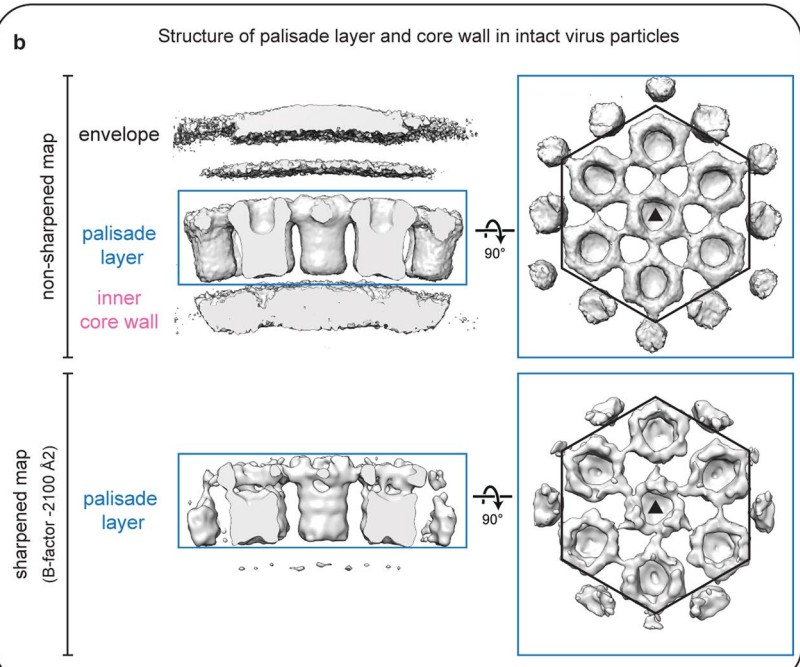

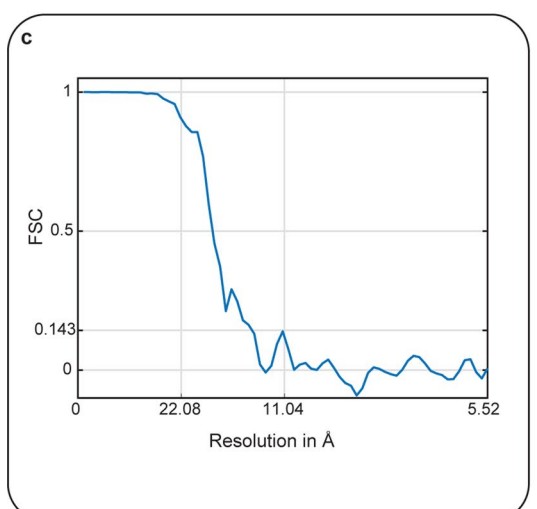

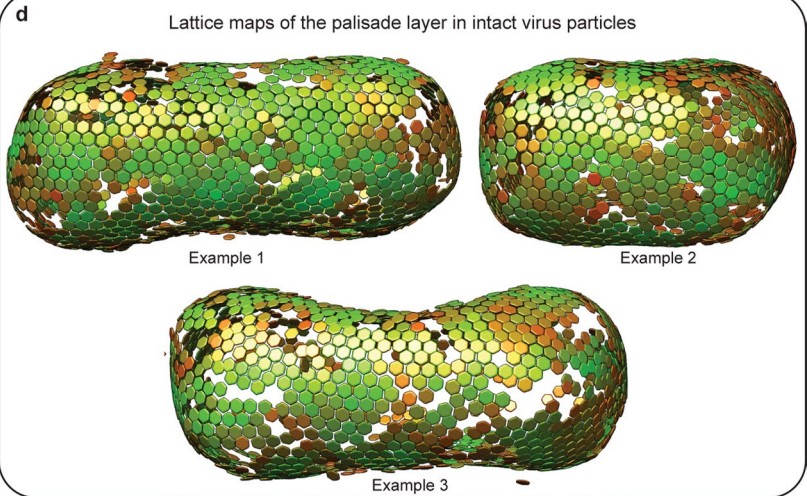

**Extended Data Fig. 4 | Subtomogram averaging of the VACV core wall in intact MV. a)** Subtomogram averaging workflow of the core in intact MV using novaCTF, Isonet and Dynamo. See Materials and Methods for more details. **b)** Non-sharpened map (top) and sharpened EM-density map (bottom) of the subtomogram average of the palisade layer and inner core wall in intact virus particles. The map is shown in a side view (left) and as seen from outside of the virus (right), where the envelope layer is removed to allow a clear view of the palisade layer. The triangle symbol annotates the central trimer and the hexagon annotates the hexamer of trimers. **c)** Fourier Shell correlation (FSC) between independent half datasets of the map shown in B. The measured resolution at the 0.143 criterion is 13.1Å. **d)** Examples of viral core lattices after final subtomogram averaging alignments. The aligned subtomogram positions are shown as hexagons to allow a facilitated interpretation of the results. Please note that each hexagon position actually represents the trimeric center of a hexamer of trimers. The color of the hexagons denotes the cross-correlation coefficient (CCC) of the alignment ranging from red (low CCC) to green (high CCC).

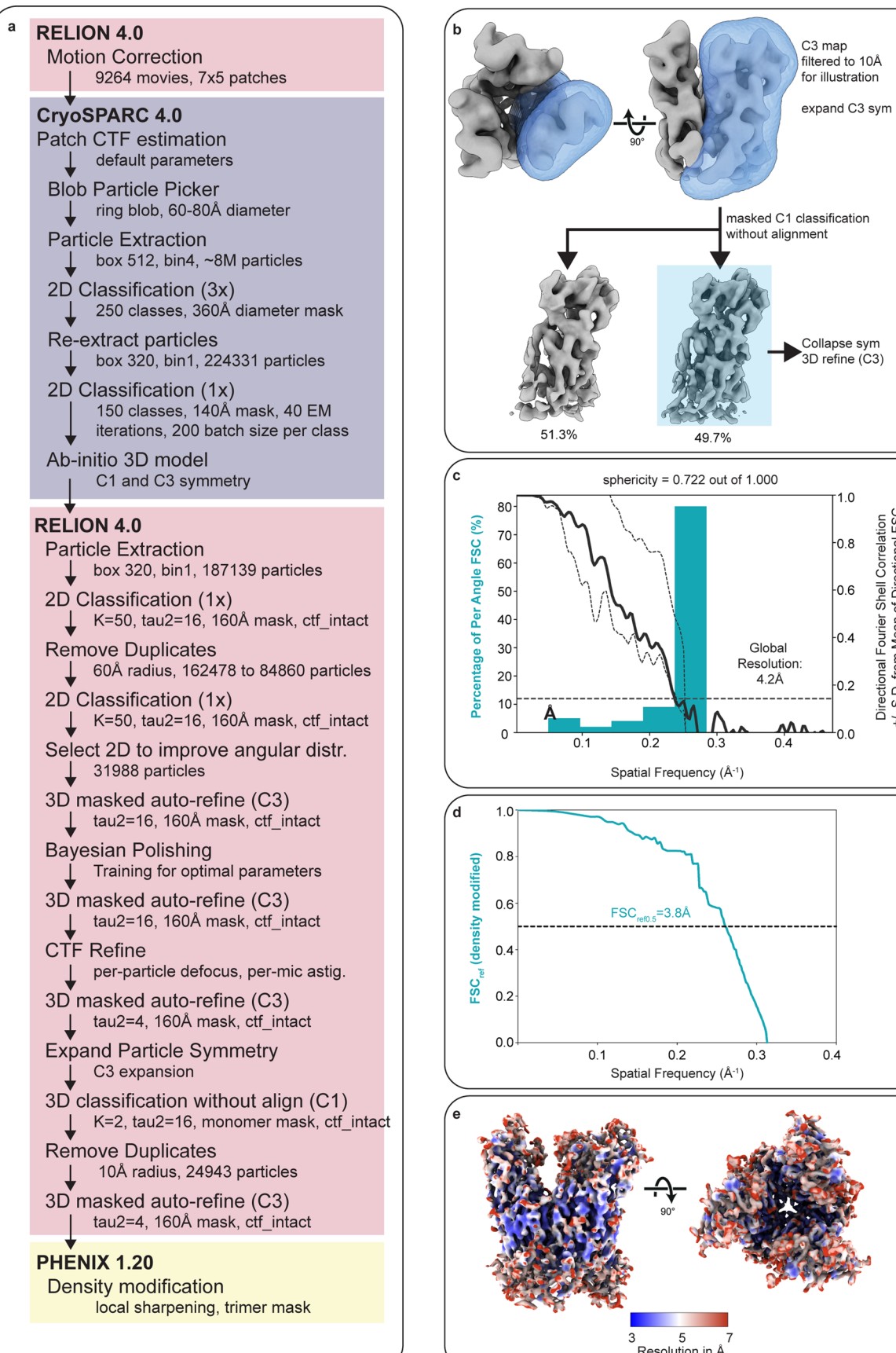

**Extended Data Fig. 5 | See next page for caption.**

**Extended Data Fig. 5 | SPA processing workflow of the A10 trimer. a)** Summary of processing steps used in the SPA workflow. **b)** 3D classification scheme used for the removal of particles containing low-quality asymmetric units within the A10 trimer. The blue mask shown was used for classification, the blue box designates which particles were selected for final refinement. **c)** 3D FSC calculations of the masked RELION[58] half maps. Cyan histogram depicts the fraction of particles that reach the corresponding resolution, and the black curves show global FSC +/− SD of FSCs calculated with extensive angular sampling. Global resolution indicated is at FSC 0.143 cutoff. **d)** $FSC_{ref}$ calculation for the Phenix density modified map with a cutoff value of 0.5 for estimated resolution. **e)** Local resolution of the A10 trimer cryo-EM density map.

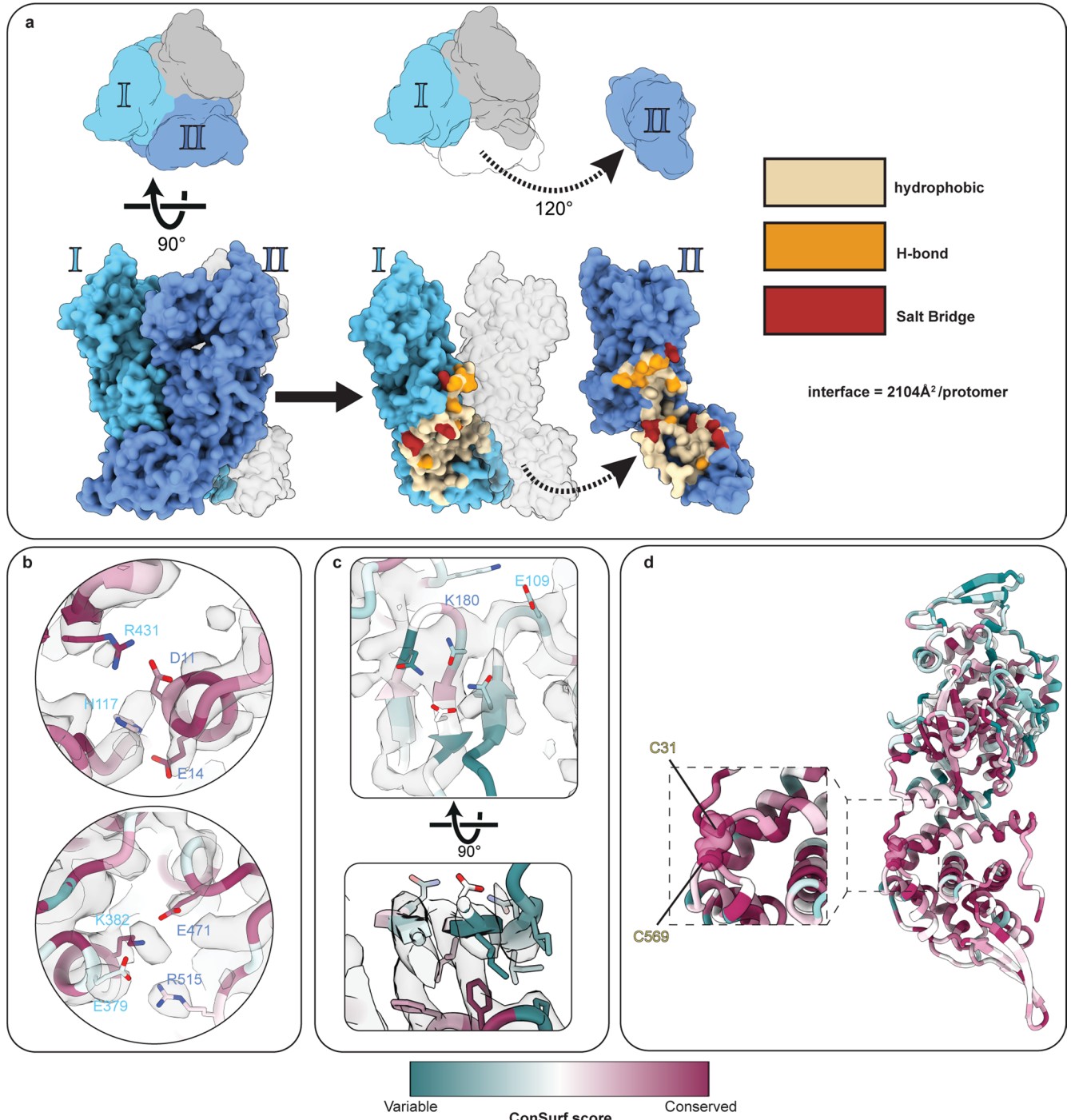

**Extended Data Fig. 6 | Inter- and intramolecular interactions of the A10 trimer. a)** Diagrammatic illustration of key inter-chain contacts at the oligomerization interface between two monomers (labeled I and II). The diagram shows monomer II pulling away from the trimer to reveal underlying contacts, shown in yellow (hydrophobic), orange (H-bond), and red (salt bridges). **b-d)** Details of the inter- and intramolecular interactions. Residues are colored according to their conservation as determined via ConSurf (see also Extended Data Fig. 3). The residue labels are colored according to the color of the chain they are in (as in panel a). **b)** Salt bridge contacts from panel A with residue numbers indicated. **c)** Top, a salt bridge and hydrogen bond network at the inter-subunit beta-sheet formed at the core of the trimer. Bottom, hydrophobic packing on the underside of the beta-sheet. Electron microscopy density is shown for all images in (b-c). **d)** Potentially interacting cysteines (residues indicated). The orientation of the monomer is identical to panel a.

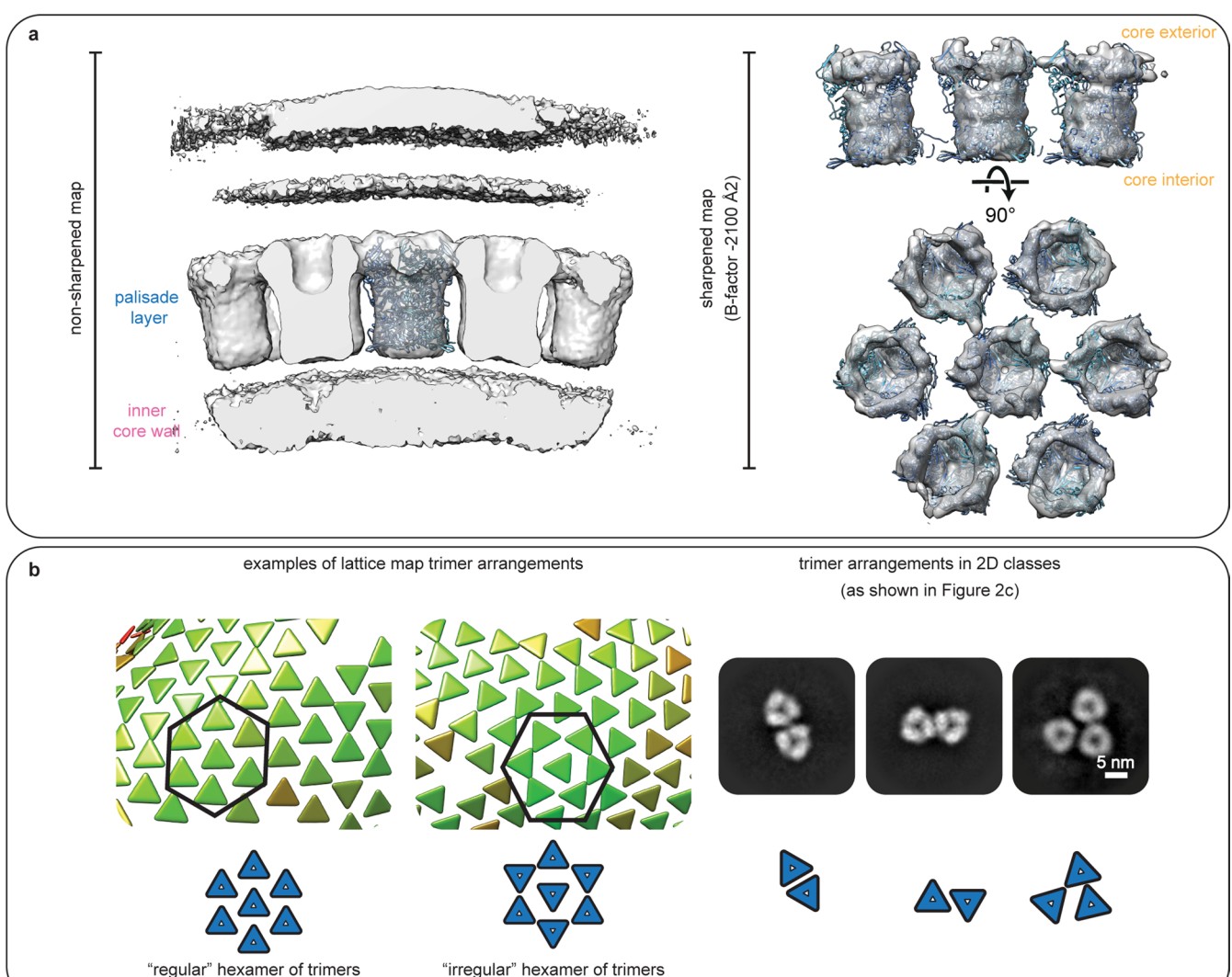

**Extended Data Fig. 7 | STA results suggest variable trimer interactions within the palisade layer. a)** Rigid body fit of the A10 trimer into the structure obtained by cryo-electron tomography and subtomogram averaging, both into the unsharpened map (left) and the sharpened map (right). In both cases, it is evident that the A10 trimer is occupying the complete density of the palisade layer. **b)** The variable positioning of the trimers with respect to each other within the palisade layer is shown in lattice maps obtained from STA of cores in intact MV (left) and in 2D classes of trimers released from the core wall of isolated cores (right). The differential orientation of the trimers to each other is always

shown schematically. We note that the variable positioning of trimers within the lattice is likely contributing to the limited resolution of our STA average. We also note that the finding of variable trimer interactions in our 2D classes supports that the variable positioning of trimers within the lattice is not due to alignment inaccuracies of STA, but rather indeed suggests the interactions of trimers among each other to be pliable. The color of the triangles denotes the cross-correlation coefficient (CCC) of the alignment ranging from red (low CCC) to green (high CCC). The 2D classes shown here are identical to the ones shown in Fig. 2c.

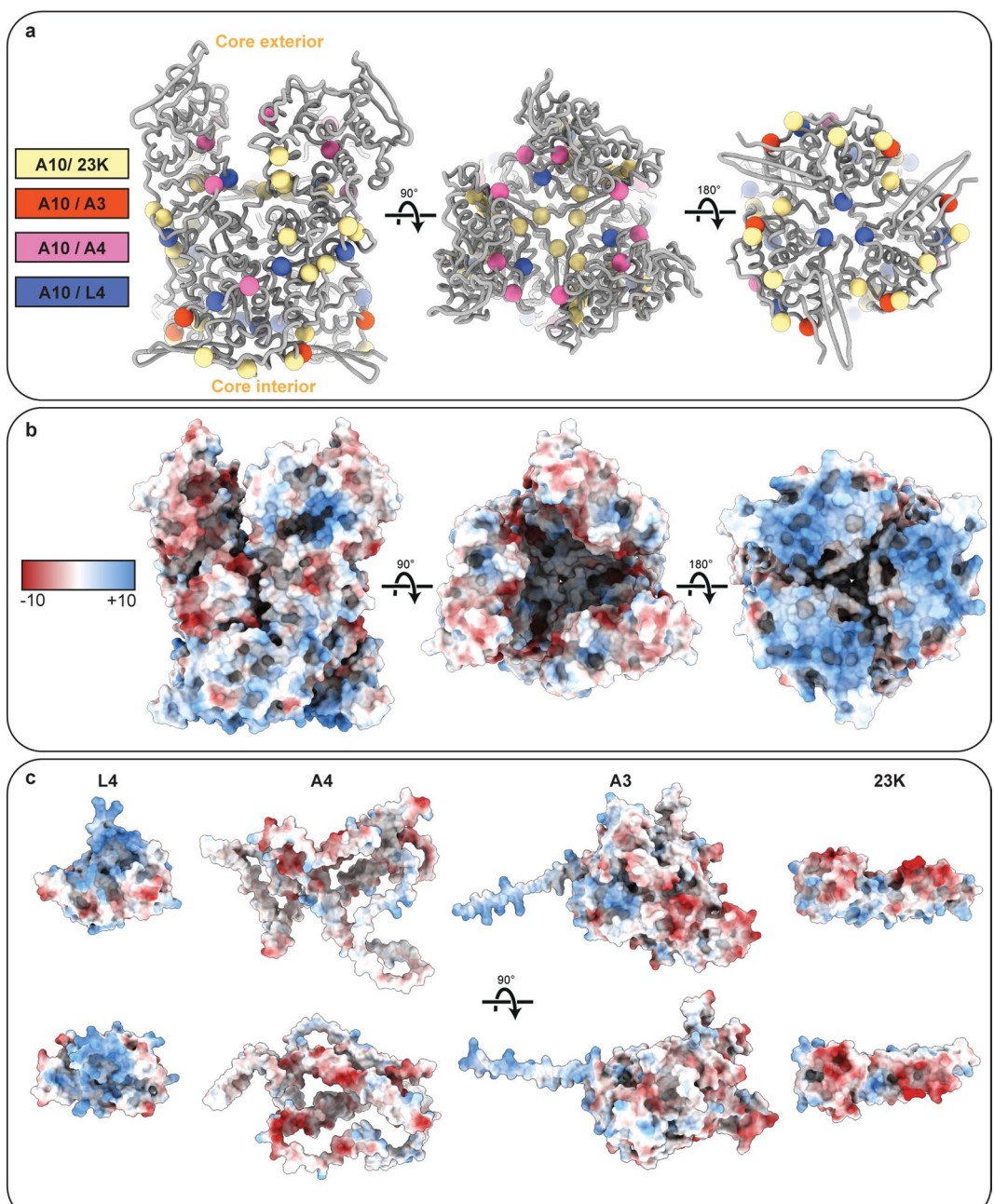

**Extended Data Fig. 8 | Cross-linking mass spectrometry sites on the A10 trimer. a)** Previously identified[19] contact sites between A10 and 23K (shown as yellow spheres), A3 (red spheres), A4 (pink spheres), and L4 (blue), as identified via cross-linking mass spectrometry. **b)** Surface charge representation of the A10 trimer (negative charge = red, positive charge = blue). **c)** Surface charge representation of AlphaFold models for monomers of L4, A4, A3 and 23K. Coloring as in (b).

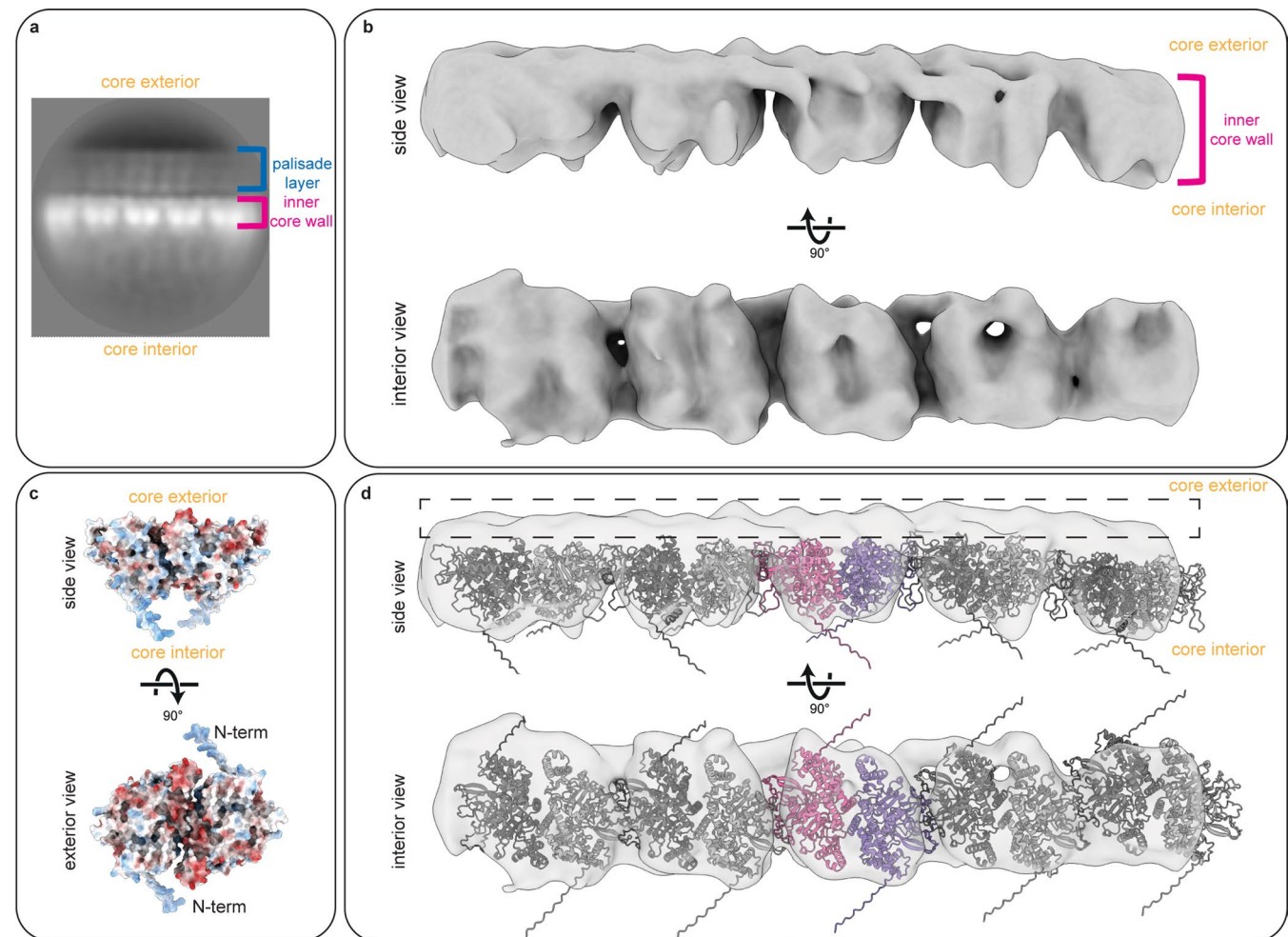

**Extended Data Fig. 9 | An A3 dimer fits to the density of the inner core wall units. a)** Projection of the 3D reconstruction of the inner core wall from panel b. **b)** Low-resolution cryo-EM reconstruction of the inner core wall as shown from side and interior views. The variability of the individual units indicates no strict long-range order, as also seen in our 2D classes (Fig. 2c) and the top view of the inner core wall in tomograms of isolated cores (Fig. 1c). **c)** AlphaFold multimer prediction of an A3 dimer (residues 62-643), with its orientation denoted relative to interior and exterior of virus core. Color represents calculated surface charge with color representations the same as in Extended Data Fig. 8. The orientation was assumed using the negative surface charge of A3 on one side and the suggested orientation towards the base of A10 with the corresponding counter charge. **d)** Pairs of A3 dimers were rigid body fit into the density of the inner core wall units in panel B. The dimensions and shape of the EM-density accommodate an A3 dimer pair. The central dimer is colored in pink and purple, neighbouring A3 pairs are coloured in grey. The black dashed rectangle annotates a small area of unoccupied density.

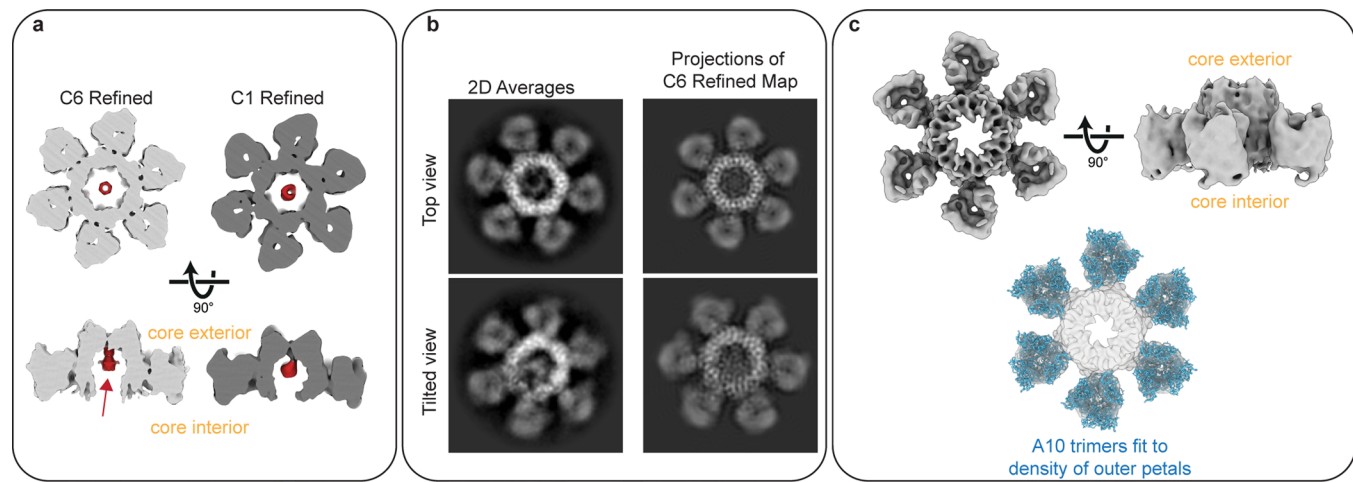

**Extended Data Fig. 10 | Single particle cryo-EM structure of the flower-shaped pore. a)** Slices through density for C6 and C1 reconstructed flower-shaped pore which reveals an unidentified, donut-shaped central density highlighted in red (red arrow). **b)** Left, Single particle cryo-EM 2D class averages of the flower-shaped pore and right, projections of the C6-symmetrized reconstruction. **c)** C6-symmetric cryo-EM reconstruction of the flower-shaped pore of the palisade layer. Bottom shows the fit of A10 trimers into the outer densities, while the central hexamer density is not modeled due to the uncertainty of its identity.

# Reporting Summary

## Statistics

For all statistical analyses, confirm that the following items are present in the figure legend, table legend, main text, or Methods section.

| n/a | Confirmed | |
|---|---|---|
| ☐ | ☒ | The exact sample size (*n*) for each experimental group/condition, given as a discrete number and unit of measurement |
| ☒ | ☐ | A statement on whether measurements were taken from distinct samples or whether the same sample was measured repeatedly |
| ☒ | ☐ | The statistical test(s) used AND whether they are one- or two-sided *Only common tests should be described solely by name; describe more complex techniques in the Methods section.* |
| ☒ | ☐ | A description of all covariates tested |
| ☒ | ☐ | A description of any assumptions or corrections, such as tests of normality and adjustment for multiple comparisons |
| ☒ | ☐ | A full description of the statistical parameters including central tendency (e.g. means) or other basic estimates (e.g. regression coefficient) AND variation (e.g. standard deviation) or associated estimates of uncertainty (e.g. confidence intervals) |
| ☒ | ☐ | For null hypothesis testing, the test statistic (e.g. *F*, *t*, *r*) with confidence intervals, effect sizes, degrees of freedom and *P* value noted *Give P values as exact values whenever suitable.* |
| ☒ | ☐ | For Bayesian analysis, information on the choice of priors and Markov chain Monte Carlo settings |
| ☒ | ☐ | For hierarchical and complex designs, identification of the appropriate level for tests and full reporting of outcomes |
| ☒ | ☐ | Estimates of effect sizes (e.g. Cohen's *d*, Pearson's *r*), indicating how they were calculated |

*Our web collection on statistics for biologists contains articles on many of the points above.*

## Software and code

Policy information about availability of computer code

| | |
|---|---|
| Data collection | Cryo-electron tomography data was collected with the SerialEM software package version 3.8. DigitalMicrograph 3.4.3 as integrated into the Gatan Microscopy Suite v3.3 (Gatan) was used for energy filter tuning. Automated single particle cryo-EM data was collected using EPU version 2.13 (Thermo Fisher Scientific) or SerialEM software package version 4.0. |
| Data analysis | Tomoman (08042020) was used to sort and create stacks. Defocus was estimated using CTFFIND4 4.1.14. IMOD (version 4.9.12) was used for tilt series alignment and to generate binned tomograms with weighted back projection. The full tomograms were reconstructed in NovaCTF (version from 2018). Missing wedge was corrected using Isonet 0.2. Subtomogram averaging was performed in Dynamo version 1.1.333. Single particle cryo-EM data were motion-corrected using the RELION 4.0-beta2 implementation of MotionCorr2 (version 4.0-beta2). Single particle cryo-EM data were further processed using cryoSPARC 4.0.0 and RELION 4.0. Volume projections were generated using EMAN version 1.9. 3D FSC calculations were done using the Salk Remote 3DFSC processing server. Protein structures were computationally predicted using Alphafold 2.3.2 and Colabfold 1.5.2. Atomic models were relaxed into cryo-EM density using Rosetta version 3.12. Atomic model was manually inspected using Coot 0.8.9.1. UCSF Chimera version 1.17.1 was used for symmetrization of atomic models. UCSF ChimeraX version 1.5 was used for RMSD calculations and visualization. Structural similarity analysis of Vaccinia virus core proteins was performed using Foldseek version 7-04e0ec8 (https://search.foldseek.com/ |

search) and Dali (accessed online August 2023) (http://ekhidna2.biocenter.helsinki.fi/dali/).
Conservation analysis was performed using ConSurf (accessed online September 2023) (https://consurf.tau.ac.il/).
Phenix version 1.20-dev-4224 was used for density modification.
Protein topology diagrams were prepared using Pro-Origami (version 1.0).
Adobe Premiere Pro 2023 was used for movie generation.
Adobe Illustrator 2023 was used for making figures.
Mass Spec data was analyzed in DIANN software version 1.8.1.

For manuscripts utilizing custom algorithms or software that are central to the research but not yet described in published literature, software must be made available to editors and reviewers. We strongly encourage code deposition in a community repository (e.g. GitHub). See the Nature Portfolio guidelines for submitting code & software for further information.

## Data

Policy information about availability of data

All manuscripts must include a data availability statement. This statement should provide the following information, where applicable:
- Accession codes, unique identifiers, or web links for publicly available datasets
- A description of any restrictions on data availability
- For clinical datasets or third party data, please ensure that the statement adheres to our policy

The electron microscopy density maps of the A10 trimer and the hexameric flower-shaped pore, the subtomogram average of the palisade layer, as well as representative tomograms for complete viruses as well as isolated cores have been deposited in the Electron Microscopy Data Bank under accession codes: EMD-17410, EMD-17411, EMD-17412, EMD-17413, EMD-17414 and EMD-18452.
The refined model of the A10 trimer has been deposited in the Protein Data Bank accession code: PDB 8P4K.
The UniProt codes of VACV core proteins used for structure prediction are: A10 (P16715), A3 (P06440), A4 (P29191), L4 (P03295).
GenBank Protein IDs of Variola virus A10 (ABF23487.1), Monkeypox virus A10 (YP_010377118.1), Rabbipox virus A10 (AAS49831.1), Cowpox virus A10 (ADZ29251.1), Ectromelia virus A10 (NP_671631.1), Orf virus P4a (AY386264.1), Amsacta moorei entomopoxvirus AMV139 (NP_064921.1), Melanoplus sanguinipes entomopoxvirus putative core protein P4a homolog (AF063866.1) were used for protein sequence alignment.
Source data for Mass Spectometry experiments is provided as zip-archived Supplementary Information with this manuscript

## Research involving human participants, their data, or biological material

Policy information about studies with human participants or human data. See also policy information about sex, gender (identity/presentation), and sexual orientation and race, ethnicity and racism.

| Reporting on sex and gender | n/a |
| Reporting on race, ethnicity, or other socially relevant groupings | n/a |
| Population characteristics | n/a |
| Recruitment | n/a |
| Ethics oversight | n/a |

Note that full information on the approval of the study protocol must also be provided in the manuscript.

## Field-specific reporting

Please select the one below that is the best fit for your research. If you are not sure, read the appropriate sections before making your selection.

☒ Life sciences  ☐ Behavioural & social sciences  ☐ Ecological, evolutionary & environmental sciences

For a reference copy of the document with all sections, see nature.com/documents/nr-reporting-summary-flat.pdf

## Life sciences study design

All studies must disclose on these points even when the disclosure is negative.

| Sample size | 9264 micrographs were collected for single particle analysis, which was sufficient to generate a high resolution density map for fitting of an atomic model. We used 15 tilt-series for subtomogram averaging, which was sufficient to generate a subtomogram average showing structural features of relevance. |
| Data exclusions | As is customary in cryo-EM data processing, entire micrographs were discarded based on poor quality metrics such as blur (poor CTF fit) or contamination. From the subset of retained micrographs, individual particles were discarded based on standard processing pipelines of 2D and 3D classification to remove particles which did not contribute valuable high resolution information. |
| Replication | Replication is not performed in macromolecular structural studies. Methods established in the field of structural biology were performed to |

| | |
|---|---|
| Replication | ensure validity of the study. |
| Randomization | Randomization of samples was not relevant in this study since structural characterization was exploratory and no grouping was necessary in order to make relevant comparisons or to draw conclusions. |
| Blinding | Blinding was not relevant in this study since macromolecular structures and assembly details were unknown to any researcher involved. |

# Reporting for specific materials, systems and methods

We require information from authors about some types of materials, experimental systems and methods used in many studies. Here, indicate whether each material, system or method listed is relevant to your study. If you are not sure if a list item applies to your research, read the appropriate section before selecting a response.

## Materials & experimental systems

| n/a | Involved in the study |
|---|---|
| ☒ | ☐ Antibodies |
| ☐ | ☒ Eukaryotic cell lines |
| ☒ | ☐ Palaeontology and archaeology |
| ☒ | ☐ Animals and other organisms |
| ☒ | ☐ Clinical data |
| ☒ | ☐ Dual use research of concern |
| ☒ | ☐ Plants |

## Methods

| n/a | Involved in the study |
|---|---|
| ☒ | ☐ ChIP-seq |
| ☒ | ☐ Flow cytometry |
| ☒ | ☐ MRI-based neuroimaging |

## Eukaryotic cell lines

Policy information about cell lines and Sex and Gender in Research

| | |
|---|---|
| Cell line source(s) | HeLa cells were kindly provided by the lab of Michael Sixt at ISTA, which were initially obtained from ATCC. |
| Authentication | not performed |
| Mycoplasma contamination | Cells were tested negative for mycoplasma contamination. |
| Commonly misidentified lines (See ICLAC register) | none |

