## [Peer Review File · Nature Structural & Molecular Biology]

Peer Review Information

Manuscript Title: Multi-modal cryo-EM reveals trimers of protein A10 to form the palisade layer in poxvirus cores

Corresponding author name(s): Florian Schur

Reviewer Comments & Decisions:

Decision Letter, initial version:
--

Message: 9th Aug 2023

Dear Dr. Schur,

Thank you again for submitting your manuscript "Multi-modal cryo-EM reveals trimers of protein A10 to form the palisade layer in poxvirus cores". We now have comments (below) from the 2 reviewers who evaluated your paper. In light of those reports, we remain interested in your study and would like to see your response to the comments of the referees, in the form of a revised manuscript.

You will see that while reviewers appreciate the results, they raise several concerns which will need to be addressed in a revision. Specifically, we ask that you explore structural and conservation analysis, suggested by reviewer #1. We agree with reviewer #2 about the deposition of inner core wall volume to EMDB.

Please be sure to address/respond to all concerns of the referees in full in a point-by-point response and highlight all changes in the revised manuscript text file. If you have comments that are intended for editors only, please include those in a separate cover letter.

We expect to see your revised manuscript within 6 weeks. If you cannot send it within this time, please contact us to discuss an extension; we would still consider your revision, provided that no similar work has been accepted for publication at NSMB or published elsewhere.

Reporting Summary:

If there are additional or modified structures presented in the final revision, please submit the corresponding PDB validation reports. In this case, and as requested by reviewers, we will ask that you submit EMDB validation reports as well.

Please note that all key data shown in the main figures as cropped gels or blots must be presented in uncropped form, with molecular weight markers. These data can be aggregated into a single supplementary figure item. While these data can be displayed in a relatively informal style, they must refer back to the relevant figures.

SOURCE DATA: we urge request that the authors provide, in tabular form, the data underlying the graphical representations used in figures. This is to further increase transparency in data reporting, as detailed in this editorial (<http://www.nature.com/nsmb/journal/v22/n10/full/nsmb.3110.html>). Spreadsheets can be submitted in excel format. Only one (1) file per figure is permitted; thus, for multi-paneled figures, the source data for each panel should be clearly labeled in the Excel file; alternately the data can be provided as multiple, clearly labeled sheets in an Excel file. When submitting files, the title field should indicate which figure the source data pertains to.

Data availability: this journal strongly supports public availability of data. All data used in accepted papers should be available via a public data repository, or alternatively, as

Supplementary Information. If data can only be shared on request, please explain why in your Data Availability Statement, and also in the correspondence with your editor. Please note that for some data types, deposition in a public repository is mandatory - more information on our data deposition policies and available repositories can be found below: <https://www.nature.com/nature-research/editorial-policies/reporting-standards#availability-of-data>

Nature Structural & Molecular Biology is committed to improving transparency in authorship. As part of our efforts in this direction, we are now requesting that all authors identified as 'corresponding author' on published papers create and link their Open Researcher and Contributor Identifier (ORCID) with their account on the Manuscript Tracking System (MTS), prior to acceptance. This applies to primary research papers only. ORCID helps the scientific community achieve unambiguous attribution of all scholarly contributions. You can create and link your ORCID from the home page of the MTS by clicking on 'Modify my Springer Nature account'. For more information please visit [visit www.springernature.com/orcid](http://www.springernature.com/orcid).

[redacted]

Sincerely,
Kat

Katarzyna Ciazynska
(she/her)
Associate Editor
Nature Structural & Molecular Biology
<https://orcid.org/0000-0002-9899-2428>

Referee expertise:

Referee #1: virology

Referee #2: virology, cryo-ET

Reviewers' Comments:

Reviewer #1:

Remarks to the Author:

A. Summary

Poxviruses have a special place in virology owing to their profound impact on human health as both the cause and potential solution to some of the most dreadful diseases. The study of these viruses led to major discoveries in cell biology. Yet, significant blindspots remain in our understanding of their biology. This manuscript presents major advances in our understanding of the poxvirus architecture and composition. It extends the pioneering cryo-ET study by Cyrklaff et al (2005) using modern approaches that prove to be particularly powerful.

I enjoyed the way the paper is written describing general features in MVs first and elucidating its components at progressively higher-resolution in subsequent sections. Key findings are that trimers of A10 form the palissade contrary to previous suggestions and do not have strong lateral interactions. This is unlike classical capsids or nucleocapsids observed in other viruses which typically feature a continuous, interconnected lattice. It suggests an unusual mechanism to achieve the brick shape typical of poxvirus cores.

The other components are identified with less confidence but the location of A3 is supported by converging evidence (published mass spectrometry and immunolabelling, and structural features in this study).

The overall model integrating these findings is novel and is likely to represent the working model for poxvirus core until higher resolution is (painstakingly) achieved.

B. Originality and significance (cf. above too)

The imaging presented in this study is a major advance compared to available data (Cyrklaff, 2005) and ingeniously combined with SPA. It changes our view of poxvirus core architecture and suggests mechanisms important for assembly.

C-D. Data, methodology and stats

State of the art approaches are used. Technical comments are added below. No major issues were identified in the validation of the SPA maps and structure.

Maps: provide local resolution maps for SPA.

Movie: add the electron density when showing A10 and the side chains involved in domain swapping.

Model building: was one chain refined and then copied to the other positions (l. 537)? It would make more sense to apply NCS restraints if that's the case. Six Ramachandran outliers remain. They should be eliminated if possible or commented on otherwise. Fig. S4 should be bigger given there's no space restriction in supplementary material

E. Conclusions

Overall, despite a limited resolution for most components the conclusions are supported by appropriate experimental evidence.

F. Suggested improvements

1. The fold of A10 should be described in more detail. Does it have any similarities with other viral or cellular proteins (Foldseek, Dali)? Even if they are models, it would also be interesting to discuss these points for A3 (and to a lesser extent L4).
2. The authors are cautious in analysing molecular details given the limited resolution of the maps. At 3.8Å resolution, the A10 trimer structure is quite reliable. An analysis of residue conservation (as implemented in ConSurf for example) would greatly support the interactions described as important for the stability of the trimer (e.g. domain swap and disulfide bond, l. 190-196) or its interaction with A3 (both interacting surfaces are expected to be conserved in addition to their complementary charges; l. 220-24 - Fig. S6).

3. A4 is discussed as a potential matrix protein. Its location, interaction on A10 trimer sides and predicted extended conformation seem more reminiscent of tape-measure or cementing proteins seen in phages (e.g. P30 of PRD1) or adenovirus (rope-like minor capsid proteins). If described as a matrix protein, hydrophobicity should be discussed.

4. Findings are extrapolated to the whole family. It is remarkable that entomopoxviruses should be overlooked here. They are the most diverse members of the family in terms of the ultrastructure and sequence, and would be the strongest support for a broad conservation of the proposed findings. If this paragraph is to stay, the findings should be at the very least extended to parapoxviruses and more divergent members of the family. Given how AF2 works, it is not surprising to see high similarities between proteins that have 95%+ sequence identity (i.e. Fig S10 is not informative).

G. References: appropriate.

H. Clarity

Overall, the manuscript is pleasant to read and the figures of high quality.

Introduction: mention EVs (line 43)

L.120: The lattice is also reminiscent of the D13 pseudo-hexagonal matrix reconstituted in vitro (Hyun, 2011 and 2022) and visualised in vivo (Heuser, 2005)

L.127-129: It would be useful to mention here that cores were treated with detergent and DTT so that the reader doesn't have to check the method section

Sucrose concentrations: w/v or v/v? 0.25% trypsin: what is this in terms of mg/ml or activity?

Reviewer #2:

Remarks to the Author:

In this manuscript, Datler et al study the structure of several components of mature vaccinia virus particles, primarily the so called palisade layer. Vaccinia virus is a poxvirus, and this family of viruses have been hard to study structurally due to their massive size (at the limit of what cryo-EM can handle) and the number of proteins that potentially make up the particle structure. The recent outbreaks of Mpox is one reason, amongst many, why it is important to better understand poxvirus architecture.

Datler et al use state-of-the-art methodology to provide important and substantial new insights into poxvirus architecture. The text is overall very well written and the figures are clearly presented and beautiful. In my opinion it would fit excellently in NSMB. I only have relatively minor comments aimed at improving the presentation of the article.

1. Since the inner core wall volume is shown in a figure, it should be deposited at EMDB. Prior to final acceptance the authors should submit a validation report.
2. With the preface that I am a structural virologist but not a poxvirologist, I wonder if the authors should also describe their particles in terms of the slightly different nomenclature that exists. In that nomenclature, which I find less ambiguous, the particles that the authors study are referred to as intracellular mature virus (IMV), distinguishing it from e.g. the extracellular enveloped virus (EEV). Perhaps it is at least worth mentioning that the particles are also referred to as intracellular mature virus (IMV)?
3. There should be a space between number and unit, for instance as "3.8 Å".
4. As a general comment, some of the supplementary material is of very high quality and could warrant being moved to main figures, if there is space.
5. line 106: The term "high-resolution cryo-ET" is ill defined. To most structural biologists, these tomograms don't count as "high-resolution", so I suggest removing that designator.
6. line 124: "average number of ~2280 trimers" Average of how many virions? What is the standard deviation?
7. line 140-142: Some statement about the inner core are made here without justifications. At least, a better visual presentation should be made, or a power spectrum shown. Alternatively, the discussion of the inner core architecture could be deferred to the place where the average is presented.
8. line 144-145: The statement "Our cryo-ET data suggested the ... trimer." should be justified if kept here. Or deferred to the discussion of the average.
9. line 182: It could be worth explicitly mentioning what part of the 614 residue A10 is not built into the density. If I understand right, it is the C-terminus with low predicted pLDDT scores?
10. line 194 and Fig. S4D: the statement about a possible disulphide should be backed up by showing the density.
11. line 209: I find this phrase very unclear: "likely represents a finding with biological significance". The authors are probably trying to say that the findings are in line with each other. The "biological significance" should be removed.
12. line 232: As far as I understand the entire central ring is unidentified. There might thus be little need to discuss the "unidentified donut-shaped density" separately (it is also very small and feature-less).
13. line 233-235: This statement relies on two assumptions which are not explicitly (quantitatively) shown: (i) that A10 dimers are shed elsewhere (it does seem to be the case), and (ii) that the central ring density without attached A10 trimers would still be particle-picked and correctly averaged. Since the authors don't explicitly show that, they

could perhaps make a weaker version of this statement, essentially saying that the trimers are still there but not suggesting that the interaction is "stronger".

14. Line 237: Even though the structure predictions make this statement probably, I would still suggest a less definitive subheading since the authors show no experimental data. As structural biologists, it is in our interest to distinguish predictions from experimental measurements. The subheading could simply say something like "is likely conserved".

15. The discussion is very long. I think it is ok that way but the readability of the article may be improved if some discussion sections were shortened a bit. Specifically, I am thinking of the subheadings "A potential role for A10..." and "Structural characterization of novel....".

16. line 252: This is nit-picking of me, but if you say you are proposing a "revised" model, that would mean that a similarly detailed other model existed, and you are proposing changes to it. I think the authors may instead want to say they are proposing a "detailed" or "more detailed" or "more accurate" model.

17. line 286: The authors say "pore hexagon" but have not provided sufficient proof that the central ring density is indeed a hexagon. Clearly, the A10 trimers around it are arranged in hexagonal fashion, but given the extent of symmetry mismatches in this virus and the limited resolution I would suggest removing the word "hexagon".

18. line 319: If I understand right, the reference to Table S1 is supposed to support the discussion of the multiple other complexes/symmetries found in the SPA data. But they are not mentioned in Table S1, or?

19. line 407: Says "Aliquots were thawed" should say "Aliquots of X were thawed" (X probably being virus...)

20. line 471: "five rounds of alignment applying no symmetry". It should be made clear what the initial alignment reference was.

21. line 480: The expression "were space cleaned (6 pixels)" is understandable only with some interpretation. I suggest to use some more standard phrase such as "overlapping particles were removed using a 6 pixel distance cutoff".

22. Fig. S3B: Are the two volumes at the top the same thing from different orientations? If so, please mark it with a rotation symbol.

23. Fig. S4: In the legend for panel C, might "inter-subunit" be a better word than "hybrid"?

24. Fig. S7C: Is the rotation arrow not in the wrong direction in panel C?

25. Fig. S7D: I realise that this is a low-resolution average and the model fitting is thus very approximate. But the authors should still mention something more about it, for instance if the subunit models also fit in other directions (or not) and possibly show such alternative fits.

26. Fig. S9A: The C6 symmetrisation of the central density appears quite speculative at this resolution. I suggest that the authors show C6 and C1 side by side already in panel A.

27. Table S1: Underfocus is typically denoted with negative values. Since the authors do that in the M&M text, I suggest doing it here too.

28. Table S2: Perhaps it would be helpful to add a column saying if the protein is a host or virus protein? For some of the proteins, that is not immediately obvious to me.

We thank the reviewers for their evaluation of our manuscript and have responded to their questions point-by-point below.

Please note that during the course of revision Lukas W Bauer has been added to the author list, which has been approved by all other authors.

We also now added one additional supplementary figure (Figure S2), which shifted the numbering of the other supplementary figures accordingly.

Reviewer #1:

Remarks to the Author:

A. Summary

Poxviruses have a special place in virology owing to their profound impact on human health as both the cause and potential solution to some of the most dreadful diseases. The study of these viruses led to major discoveries in cell biology. Yet, significant blindspots remain in our understanding of their biology. This manuscript presents major advances in our understanding of the poxvirus architecture and composition. It extends the pioneering cryo-ET study by Cyrklaff et al (2005) using modern approaches that prove to be particularly powerful.

I enjoyed the way the paper is written describing general features in MVs first and elucidating its components at progressively higher-resolution in subsequent sections. Key findings are that trimers of A10 form the palissade contrary to previous suggestions and do not have strong lateral interactions. This is unlike classical capsids or nucleocapsids observed in other viruses which typically feature a continuous, interconnected lattice. It suggests an unusual mechanism to achieve the brick shape typical of poxvirus cores.

The other components are identified with less confidence but the location of A3 is supported by converging evidence (published mass spectrometry and immunolabelling, and structural features in this study).

The overall model integrating these findings is novel and is likely to represent the working model for poxvirus core until higher resolution is (painstakingly) achieved.

We thank the reviewer for the positive comments.

B. Originality and significance (cf. above too)

The imaging presented in this study is a major advance compared to available data (Cyrklaff, 2005) and ingeniously combined with SPA. It changes our view of poxvirus core architecture and suggests mechanisms important for assembly.

C-D. Data, methodology and stats

State of the art approaches are used. Technical comments are added below. No major issues were identified in the validation of the SPA maps and structure.

Maps: provide local resolution maps for SPA.

We now provide local resolution maps and show them as new panels in Figure S4 and S9.

Movie: add the electron density when showing A10 and the side chains involved in domain swapping.

We acknowledge the reviewer's comment. However, we have refrained from adding the electron density to the movie as this made the information difficult to interpret, given that for some of the smaller and negatively charged side chains no clear density is visible. This information is already stated

in the manuscript: *“We note that at ~4 Å resolution structural features, such as small and/or negatively charged side chains are not clearly visible and hence impose a limit to interpretability.”*

As the corresponding EM density map and model are deposited to the EMDB/PDB, interested readers can inspect the map and model in more detail than what we would be able to convey in the video

Model building: was one chain refined and then copied to the other positions (l. 537)? It would make more sense to apply NCS restraints if that’s the case. Six Ramachandran outliers remain. They should be eliminated if possible or commented on otherwise.

We agree with the reviewer and we have now performed a final iteration of Rosetta refinement into the cryo-EM density with NCS applied to the model. The backbone RMSD between our new model and the previous one without NCS is 0.873Å for 563 pruned atom pairs, and 1.065Å across all 599 pairs. Molprobit statistics are improved compared to our previous model, and there are no longer Ramchandran outliers. We thank the reviewer for their suggestion. We have updated the model statistics table (Table S3) and the methods section to include the Rosetta refinement with NCS restraints. We have also updated the figures and movies with the new model where necessary.

Fig. S4 should be bigger given there’s no space restriction in supplementary material
We have changed the layout of Figure S4 to increase its size.

E. Conclusions

Overall, despite a limited resolution for most components the conclusions are supported by appropriate experimental evidence.

F. Suggested improvements

1. The fold of A10 should be described in more detail. Does it have any similarities with other viral or cellular proteins (Foldseek, Dali)? Even if they are models, it would also be interesting to discuss these points for A3 (and to a lesser extent L4).

We thank the reviewer for these suggestions. We have now performed a Dali analysis of A10, which revealed no convincing similarities to any other proteins. We repeated this search using Foldseek against the PDB or AlphaFold-Database, again obtaining the same result. For example, for A10 Foldseek reports only hits with small protein stretches, all with high E-value (where lower is better) and low TM-score (where higher would be better).

In the case of A3, Foldseek and Dali reveal similarities in its protein fold to deubiquinating enzymes, such as ubiquitin carboxyl-terminal hydrolases. This is very interesting and potentially warrants further analysis, but currently goes beyond the scope of this manuscript.

Similarly, for L4 and 23K the Dali and Foldseek hits were not providing convincing results with high scores.

We now briefly mention these results in the main text (changes in bold).

“No experimentally-derived structures of the major structural core protein candidates are available. Hence, we used AlphaFold (Jumper et al. 2021) to computationally predict models of the main core protein candidates, A10, 23K, A3, A4, and L4 (Figure S1C) to facilitate the interpretation of cryo-EM densities obtained in our downstream workflow. **Foldseek (van Kempen et al. 2023) and Dali (Holm 2022) analysis revealed no similarities in protein fold of A10 to other cellular or viral proteins in the Protein Data Bank (PDB) or AlphaFold-databases. Interestingly, A3 showed highest similarity to deubiquinating proteins. For the other proteins Foldseek and Dali hits had low probability or TM**

scores, further highlighting the structural dissimilarity of these proteins to cellular proteins or other viral proteins outside of the poxvirus family.”

The Foldseek analysis of A4 is explained below, when answering comment #3 about the role of A4 in the viral core.

2. The authors are cautious in analysing molecular details given the limited resolution of the maps. At 3.8Å resolution, the A10 trimer structure is quite reliable. An analysis of residue conservation (as implemented in ConSurf for example) would greatly support the interactions described as important for the stability of the trimer (e.g. domain swap and disulfide bond, l. 190-196) or its interaction with A3 (both interacting surfaces are expected to be conserved in addition to their complementary charges; l. 220-24 - Fig. S6).

The reviewer is correct that given the high sequence conservation of the A10 protein within the orthopoxvirus genus (Figure S11A in the revised manuscript), the role of the discussed amino acids in forming interactions is somewhat obvious, and hence results in the same AF2 predictions.

As suggested by the reviewer, we now have performed ConSurf-analysis of A10, which looks at sequence conservation beyond the orthopoxvirus genus. Both cysteine residues (C31/C569) in A10 are highly conserved according to our ConSurf analysis. Similarly, the salt bridges described to tether central helices together within the trimer are also showing high conservation. Interestingly, the residues stretching the region forming the three-stranded beta-sheet show an average to variable conservation. However, it is notable that these residues are conserved within orthopoxviruses, but less so among all other poxviruses included in ConSurf analysis (see also our comparison to parapoxviruses and entomopoxviruses below).

We have also extended this ConSurf analysis to the other structural core proteins and provide these results in a new Figure S2 and also an updated Figure S5 (previously Figure S4), which displays the conservation of residues plotted back onto the 3D model.

The base of the A10 trimer is more strongly conserved than its top. This includes several of the positively charged residues which have an average to high conservation scale according to ConSurf. The increased conservation of the base of A10 is interesting given the similarity of structure predictions for this region among orthopox, parapox and entomopoxviruses (see please our answer to comment #4 below).

The side of A3 we predict to be contacting the base of the A10 trimer also shows an average level of conservation, where several negatively charged residues are conserved. This supports our speculation about how A10 and A3 could be in contact with each other.

However, given that based on our structures we are not able to unambiguously define potential residue-residue contacts (or also exclude the presence of an additional protein in between the trimer and inner core wall, as discussed in our manuscript could be 23K), we prefer not to speculate in too much detail about conservation of a specific interface.

New Figure S2 below:

ConSurf Scoring of major structural core proteins

Figure S2: Sequence conservation analysis of structural core proteins

AlphaFold-predicted models of A10, A3, 23K, A4 and L4, shown in the same orientation as in Figure S1C and colored according to their ConSurf score²⁹, highlighting the predicted conservation of individual residues. The modeled residues are indicated in brackets.

Updated Figure S5 below:

Figure S5: Inter- and intramolecular interactions of the A10 trimer

A) Diagrammatic illustration of key inter-chain contacts at the oligomerization interface between two monomers (labeled I and II). The diagram shows monomer II pulling away from the trimer to reveal underlying contacts, shown in yellow (hydrophobic), orange (H-bond), and red (salt bridges). **B-D)** Details of the inter- and intramolecular interactions. Residues are colored according to their conservation as determined via ConSurf (see also Figure 2). The residue labels are colored according to the color of the chain they are in (as in panel A). **B)** salt bridge contacts from panel A with residue numbers indicated. **C)** Top, a salt bridge and hydrogen bond network at the inter-subunit beta-sheet formed at the core of the trimer. Bottom, hydrophobic packing on the underside of the beta-sheet. **D)** intramolecular disulfide (cysteine residues indicated). The orientation of the monomer is identical to panel A.

4. Findings are extrapolated to the whole family. It is remarkable that entomopoxviruses should be overlooked here. They are the most diverse members of the family in terms of the ultrastructure and sequence, and would be the strongest support for a broad conservation of the proposed findings. If this paragraph is to stay, the findings should be at the very least extended to parapoxviruses and more divergent members of the family. Given how AF2 works, it is not surprising to see high similarities between proteins that have 95%+ sequence identity (i.e. Fig S10 is not informative).

Given that comment #2 and #4 deal both with conservation of residue and structure, we decided to answer #4 before providing answers to comment #3.

The reviewer is correct that our extrapolation of the role and structure of the A10 trimer to the entire poxvirus family does not include potential differences in more diverse members. Hence, we have now extended our analysis of the A10 protein structure conservation section to also include statements about Entomopoxviruses and Parapoxviruses and extended Figure S11 with this additional information.

The A10 homolog in Orf virus (a member of the *Parapoxvirus* genus), having only ~40% sequence identity, resulted in an AF-prediction almost identical to what we observe in orthopoxviruses. However, the fold of the heterodimeric three-stranded beta-sheet is slightly different compared to as seen in orthopoxviruses. Despite this small difference, given the overall identical trimer prediction for Orf virus, it is still fair to assume that the trimer structure is likely conserved among both ortho- and parapoxviruses.

We have also extended our comparison to Entomopoxviruses. Specifically, we performed a sequence alignment between protein A10 of VACV and the respective homologs in *Amsacta moorei* entomopoxvirus (AmEPV) and *Melanoplus sanguinipes* entomopoxvirus (MSEPV). Here the sequence conservation was substantially lower with ~22%. For these two viruses, AlphaFold predicted an overall different protein structure for the protein A10 homologs.

However, a region in these entomopoxvirus proteins adopted an almost identical conformation compared to the base of the A10 trimer in orthopoxviruses, Specifically, AmEPV residues 618-850 and MSEPV residues 784-1014 adopt a fold that is structurally remarkably similar with VACV protein A10 residues 370-602.

This is intriguing as it suggests that the region in A10 that is positioned towards the inner core wall might be a defining structural part among different more distantly related poxvirus species.

To illustrate all these comparisons, we have now extended the table shown in **Figure S11A** with the sequence conservation of Orf virus, AmEPV and MSEPV. We have also included the AF-prediction of Orf virus into Figure S11 panel B and created a new Figure S11 panel C for the entomopoxvirus predictions.

Further, we changed the corresponding section title to “The A10 trimer is likely conserved across ortho- and parapoxvirus species” and added there the results obtained for the other poxvirus members.

(changes in bold): “The protein sequence of A10 is highly conserved among the orthopoxvirus genus, including the key residues forming interactions in the trimer, with an average ~97% sequence identity between VACV Western reserve (WR), Variola virus, Monkeypox virus, Rabbitpox virus, Cowpox virus and Ectromelia virus (Figure S11A). Correspondingly, AlphaFold predictions of these proteins as monomers or trimers are very similar (Figure S11B).

More distantly related poxviruses show a substantially lower sequence identity of their A10 protein homologs, with ~40% identity in Orf virus (a parapoxvirus) and ~22% in two members of the entomopoxvirus genus, *Amsacta moorei* entomopoxvirus (AmEPV) and *Melanoplus sanguinipes*

entomopoxvirus (MSEPV) (Figure S11A). AlphaFold predictions for Orf virus showed a similar A10 trimer structure (Figure S11B). AmEPV and MSEPV protein A10 homologs displayed a different overall fold. However, one region in AeMV and MSEPV A10 homologs adopted a similar conformation compared to the base of A10 trimer in orthopoxviruses (Figure S11C). This is intriguing as it suggests that the region in A10 that is positioned towards the inner core wall might be a defining structural part among different poxvirus species.

Overall, these findings indicate that the interactions formed within the trimers constituting the palisade layer are similar among members of the orthopox and parapoxvirus genus, but potentially display interesting differences to entomopoxviruses.”

New Figure S11 below:

Figure S11: Comparison of Vaccinia virus Western Reserve A10 trimers to other members of the poxvirus family

A) Sequence identity of A10 protein of different viruses of the poxvirus family in comparison to VACV WR. **B)** Comparison of the initial VACV WR A10 AlphaFold (AF2) prediction to the refined VACV WR A10 structure (also shown in Figure 3) and AF2 predictions of Variola virus A10 and the parapoxvirus Orf virus P4a (A10) residues 1-599. This comparison shows the strong similarity between the protein folds between these virus species. It further shows that the biggest difference between the predicted and refined VACV WR A10 model is at the top of the trimer, facing the outside of the core. The color code displays RMSD (root-mean-square deviation) variations, with lower values equaling a more similar structure. **C)** Comparison of the refined VACV A10 model to the predicted putative core protein models

of MSEPV and AmEPV. Analysis revealed that despite an overall more different fold, MSEPV residues 784-1014 and AmEPV residues 618-850 adopt a highly similar fold compared to VACV A10 residues 370-599. Please note that in the AmEPV AF2-prediction a slightly different angle in a connecting loop between two halves of the fold (annotated with an arrow) leads to a different orientation.

3. A4 is discussed as a potential matrix protein. Its location, interaction on A10 trimer sides and predicted extended conformation seem more reminiscent of tape-measure or cementing proteins seen in phages (e.g. P30 of PRD1) or adenovirus (rope-like minor capsid proteins). If described as a matrix protein, hydrophobicity should be discussed.

This is an excellent point and indeed the analogy of A4 to cement proteins in Adenoviruses is appealing. Interestingly, when using the FoldSeek server, the only hit for a protein matching A4 within the PDB is for Minor/Cement protein IX of human Adenovirus (hAdV) 5, as reported in pdb 6B1T. Specifically, the C-terminal helix of A4 (which is also more highly conserved than the rest of the protein, see new Figure S2) matches the C-terminal helix of protein IX, which is involved in a 4-helix bundle. However, this match has a high E-value (9.29e+0) and low TM-score (0.344), which is suboptimal.

There were only two other matches in the AFDB50 for two uncharacterized proteins. One in the barn swallow (*Hirundo rustica*), and the other in the rainbow trout (*Oncorhynchus mykiss*), both with high E-values and low TM-scores.

Hence, while the match to hADV5 protein IX is very interesting, it is also still a preliminary analysis. Therefore, we refrain to discuss this in too much detail in our manuscript. Instead, we now briefly mention in the discussion that the role of A4 as matrix protein would be reminiscent of cement proteins in Adenoviruses, also based on their extended conformation.

With our conclusion that A4 could act as a matrix-like protein, we have referred to the report by Cudmore et al. 1996, who characterized the membrane-binding features of A4 (or p39 called in their manuscript) in great detail. Specifically, they reported A4 to be of predominantly hydrophilic nature and not to bind directly to membrane. Instead, they concluded that A4 would be positioned to link the core with surrounding membranes by interacting with other viral membrane proteins. We acknowledge that our statement referencing this study was not concise enough and hence modified this section to clarify this aspect.

“The extended conformation of A4 is reminiscent to minor coat proteins found in Adenoviruses, termed cement proteins due to their role in assembling and maintaining the virus shell (Dai et al. 2017, Gallardo et al. 2021). Hence, A4 could have a similar role compared to these proteins in providing additional stabilization to the palisade layer. A4 has previously been also described as matrix-like protein, which for example could establish a link between the core and surrounding membranes via binding other viral membrane proteins (Cudmore et al. 1996).”

G. References: appropriate.

H. Clarity

Overall, the manuscript is pleasant to read and the figures of high quality.
Thank you!

Introduction: mention EVs (line 43)
We have done this as suggested.

L.120: The lattice is also reminiscent of the D13 pseudo-hexagonal matrix reconstituted in vitro (Hyun, 2011 and 2022) and visualised in vivo (Heuser, 2005)

This statement and the respective references have now been added.

L.127-129: It would be useful to mention here that cores were treated with detergent and DTT so that the reader doesn't have to check the method section

We have added this information.

“In order to improve the resolution of core structural features, we decided to reduce the complexity of our experimental system and therefore isolated VACV cores via optimizing established protocols using the detergent NP40 and dithiothreitol (DTT) (Joklik 1962, Esteban 1984, Dubochet et al. 1994).

Sucrose concentrations: w/v or v/v? 0.25% trypsin: what is this in terms of mg/ml or activity?

We now specify in the methods that the sucrose concentrations are w/v.

For Trypsin, we use a commercially available 0.25% Trypsin solution (Thermo Fisher Scientific, no. 25200056). Based on the manufacturer’s information available on their website, the concentration of the individual components is the following:

Components	Molecular Weight	Concentration (mg/L)	mM
Inorganic Salts			
Potassium Chloride (KCl)	75.0	400.0	5.3333335
Potassium Phosphate monobasic (KH ₂ PO ₄)	136.0	60.0	0.44117647
Sodium Bicarbonate (NaHCO ₃)	84.0	350.0	4.1666665
Sodium Chloride (NaCl)	58.0	8000.0	137.93103
Sodium Phosphate dibasic (Na ₂ HPO ₄ ·7H ₂ O)	268.0	90.0	0.33582088
Other Components			
D-Glucose (Dextrose)	180.0	1000.0	5.5555553
EDTA 4Na 2H ₂ O	416.2	380.0	0.9130226
Phenol Red	398.0	10.0	0.025125628
Trypsin	23800.0	2500.0	0.10504202

Obtained from <https://www.thermofisher.com/at/en/home/technical-resources/media-formulation.298.html>, accessed on August 21.

Reviewer #2:

Remarks to the Author:

In this manuscript, Datler et al study the structure of several components of mature vaccinia virus particles, primarily the so called palisade layer. Vaccinia virus is a poxvirus, and this family of viruses have been hard to study structurally due to their massive size (at the limit of what cryo-EM can handle) and the number of proteins that potentially make up the particle structure. The recent outbreaks of Mpox is one reason, amongst many, why it is important to better understand poxvirus architecture.

Datler et al use state-of-the-art methodology to provide important and substantial new insights into poxvirus architecture. The text is overall very well written and the figures are clearly presented and beautiful. In my opinion it would fit excellently in NSMB. I only have relatively minor comments aimed at improving the presentation of the article.

We thank the reviewer for the encouraging comments.

1. Since the inner core wall volume is shown in a figure, it should be deposited at EMDB. Prior to final acceptance the authors should submit a validation report.

The volume for the inner core wall has been deposited to the EMDB under deposition code EMD-18452. A validation report is now provided with the revision.

2. With the preface that I am a structural virologist but not a poxvirologist, I wonder if the authors should also describe their particles in terms of the slightly different nomenclature that exists. In that nomenclature, which I find less ambiguous, the particles that the authors study are referred to as intracellular mature virus (IMV), distinguishing it from e.g. the extracellular enveloped virus (EEV). Perhaps it is at least worth mentioning that the particles are also referred to as intracellular mature virus (IMV)?

We have changed this section, also following the recommendation of reviewer 1, and now distinguish between intracellular mature virus (MV) and extracellular enveloped virus (EV). We agree with reviewer 2 that the usage of the abbreviation IMV, instead of just using MV, is less ambiguous. However, we have aimed to be consistent also with the accompanying manuscript by Liu & Corroyer-Dulmot *et al.*, and also with the current nomenclature in the poxvirus field.

3. There should be a space between number and unit, for instance as “3.8 Å”.

We have inserted a space between a number and the Å symbol, for each instance where it occurs.

4. As a general comment, some of the supplementary material is of very high quality and could warrant being moved to main figures, if there is space.

We thank the reviewer for this nice comment. However, due to space reasons we have refrained from moving supplementary figures to main figures.

5. line 106: The term “high-resolution cryo-ET” is ill defined. To most structural biologists, these tomograms don’t count as “high-resolution”, so I suggest removing that designator.

We have removed the term “*high-resolution*” as suggested.

6. line 124: “average number of ~2280 trimers” Average of how many virions? What is the standard deviation?

The average number was calculated from 15 virions. The standard deviation is ± 309 .

We now provide these numbers also in the revised manuscript (see below, changes in bold).

*“Using the initial mesh defined on the surface of the core wall for extracting subtomograms, and the measured size of a trimer within a hexamer of trimer unit, we calculated an average number of ~2,280 trimers (**standard deviation (SD) = ± 309 , n = 15 virions**) to constitute the palisade layer, not considering the presence of gaps and cracks.”*

7. line 140-142: Some statement about the inner core are made here without justifications. At least, a better visual presentation should be made, or a power spectrum shown. Alternatively, the discussion of the inner core architecture could be deferred to the place where the average is presented.

We realized that our statement about the inner core at this position in the text was not clear. Hence, we have changed this section to make it more explicit how we have obtained the measurements of the inner core wall units.

Specifically, we believe that in Figure 1C the shape of the inner core wall units is already easily appreciated. Together with the Movie S2, where the organization of the inner core wall is also visible, we are confident that our statements about the inner core wall unit at this position of the manuscript are warranted.

“More importantly, we could for the first time observe the structural arrangement of the inner core wall (Figure 1C, inner core wall panel, see also Movie S2). Each inner core wall unit adopted a square-like shape of $\sim 7.4\text{nm} \times 7.4\text{nm}$ (as measured directly from tomogram slices containing the inner core wall units) and appeared to be following at least 2-fold symmetry, which is consistent with our interpretation from intact viruses that the inner core wall is not following the organization of the palisade layer.”

8. line 144-145: The statement “Our cryo-ET data suggested the ... trimer.” should be justified if kept here. Or deferred to the discussion of the average.

We have removed this sentence.

9. line 182: It could be worth explicitly mentioning what part of the 614 residue A10 is not built into the density. If I understand right, it is the C-terminus with low predicted pLDDT scores?

We now state in the methods and Table S3 which residues are built into the EM-density (residues 1-599).

10. line 194 and Fig. S4D: the statement about a possible disulphide should be backed up by showing the density.

We acknowledge that we have been not precise in this statement in the initial version of the manuscript.

We made the conclusion that a disulfide bridge could exist based on the proximity of the two cysteines in our model. However, we did not actually observe a clear density for it, which is also reasonable given that we purified virus cores under reducing conditions. We still found it relevant to mention these cysteines, considering a study that suggested that within assembled MVs, core proteins including A10 are able to form disulfide bonds that are required for maintaining stability of released virus particles (Locker and Griffiths 1999, DOI: [10.1083/jcb.144.2.267](https://doi.org/10.1083/jcb.144.2.267)). Also, given that the two cysteines are close to each other in space, but not sequence, they might have a role defining protein conformation. We have now changed this part of the manuscript to make this clearer.

“A previous study suggested that VACV core proteins form disulfide bonds within MVs relevant for maintaining stability of released virus particles (Locker and Griffiths 1999). When analysing our A10 model, we found two highly conserved cysteine residues (C31 and C569) in close proximity to each other (Figure S5D) which conceivably could clamp the N-terminal and C-terminal parts of A10 monomer together, stabilizing its conformation.”

We provide information about the conservation of cysteines already in our response to reviewer 1.

11. line 209: I find this phrase very unclear: “likely represents a finding with biological significance”. The authors are probably trying to say that the findings are in line with each other. The “biological significance” should be removed.

We noticed that the sentence containing this phrase was ambiguous in its meaning and we have therefore removed it.

12. line 232: As far as I understand the entire central ring is unidentified. There might thus be little need to discuss the “unidentified donut-shaped density” separately (it is also very small and featureless).

We have removed the sentence mentioning the unidentified donut-shaped density in the main text. However, we have kept the reference mentioning it in the legend of Figure S10.

13. line 233-235: This statement relies on two assumptions which are not explicitly (quantitatively) shown: (i) that A10 dimers are shed elsewhere (it does seem to be the case), and (ii) that the central ring density without attached A10 trimers would still be particle-picked and correctly averaged. Since the authors don't explicitly show that, they could perhaps make a weaker version of this statement, essentially saying that the trimers are still there but not suggesting that the interaction is “stronger”.

The reviewer is correct, we have not quantitatively analysed shedding and how efficiently we can detect the central ring density when no trimers are attached. Hence, we have now removed the entire sentence.

14. Line 237: Even though the structure predictions make this statement probably, I would still suggest a less definitive subheading since the authors show no experimental data. As structural biologists, it is in our interest to distinguish predictions from experimental measurements. The subheading could simply say something like “is likely conserved”.

We have now changed the subheading and also extended the rest of the section based on comments of reviewer 1 (please see our answer above).

15. The discussion is very long. I think it is ok that way but the readability of the article may be improved if some discussion sections were shortened a bit. Specifically, I am thinking of the subheadings “A potential role for A10...” and “Structural characterization of novel....”.

We have removed one sentence in each of the mentioned sections, given that they were not absolutely necessary. We have also shortened the very first paragraph of the discussion.

16. line 252: This is nit-picking of me, but if you say you are proposing a “revised” model, that would mean that a similarly detailed other model existed, and you are proposing changes to it. I think the authors may instead want to say they are proposing a “detailed” or “more detailed” or “more accurate” model.

As suggested by the reviewer, we have changed the phrase “revised model” to “more detailed model” in the discussion and also the abstract.

17. line 286: The authors say “pore hexagon” but have not provided sufficient proof that the central ring density is indeed a hexagon. Clearly, the A10 trimers around it are arranged in hexagonal fashion, but given the extent of symmetry mismatches in this virus and the limited resolution I would suggest removing the word “hexagon”.

We have removed the word “hexagon”.

18. line 319: If I understand right, the reference to Table S1 is supposed to support the discussion of the multiple other complexes/symmetries found in the SPA data. But they are not mentioned in Table S1, or?

We thank the reviewer for noticing this error and have removed the reference to Table S1 and this position.

19. line 407: Says “Aliquots were thawed” should say “Aliquots of X were thawed” (X probably being virus...)

We have changed the statement to now read:
“Aliquots **of isolated cores** were thawed.....”

20. line 471: “five rounds of alignment applying no symmetry”. It should be made clear what the initial alignment reference was.

We now explain how the initial reference was obtained (see below, changes in bold).

*“To generate a de novo reference, subtomograms (cubic size 464 Å³) were then extracted from IsoNet-corrected bin8 tomograms and subjected to five rounds of alignment applying no symmetry. **The first initial alignment reference was generated by averaging all particles, using their non-refined starting positions.**”*

21. line 480: The expression “were space cleaned (6 pixels)” is understandable only with some interpretation. I suggest to use some more standard phrase such as “overlapping particles were removed using a 6 pixel distance cutoff”.

We have changed the expression as suggested (see below, changes in bold).

*“After the first two alignments in bin8, **overlapping particles were removed using a 6 pixel distance cutoff** and.....”*

22. Fig. S3B: Are the two volumes at the top the same thing from different orientations? If so, please mark it with a rotation symbol.

Yes, the volumes on top are identical, but rotated by 90 degrees. We have added rotation symbol to illustrate this.

23. Fig. S4: In the legend for panel C, might “inter-subunit” be a better word than “hybrid”?

We have changed the word “hybrid” to “inter-subunit”.

24. Fig. S7C: Is the rotation arrow not in the wrong direction in panel C?

We thank the reviewer for noticing this error. We have corrected the orientation of the arrow.

25. Fig. S7D: I realise that this is a low-resolution average and the model fitting is thus very approximate. But the authors should still mention something more about it, for instance if the subunit models also fit in other directions (or not) and possibly show such alternative fits.

We acknowledge that the current map is limited in resolution and is anisotropic, preventing the possibility to unambiguously assign map handedness and orientation of an A3 dimer. We now more explicitly acknowledge these limitations within the text (see below).

Nonetheless, we have attempted fits of the A3 dimer model into our volume with and without flipping handedness (see figure R1, below), and found that the current fit has the highest correlation value.

For fitting the highest ranking Alphafold2 model of A3 dimer into the inner core density we rigid body fit using chimera's "fit in map" tool. Options were set for real-time correlation using a simulated map at resolution at 20 Å (estimated resolution from cryoSPARC) and optimization by correlation. We fit the dimer in all possible orientations and, as expected with low resolution volumes, correlation values were similar for all fits although our current fit resulted in the highest value. The current orientation also filled the innermost density most completely, while maintaining an apparent two-fold symmetry (as appears in 2D class averages). We provide a figure for the reviewer to allow a better appreciation of these fits.

We have modified the methods to describe how we carried out fitting, and modified the main text to acknowledge that alternative configurations are possible:

“Although the low resolution and anisotropy of this map prevented determination of map handedness and unambiguously assigning orientation of A3 within the volume, our model with A3’s negatively charged patch facing towards A10 satisfied the inner density most completely.”

Figure R1. Possible fits of alpha-fold A3 dimer model into density for inner core wall volume with and without map handedness flipped (figure not included in the revised manuscript).

26. Fig. S9A: The C6 symmetrisation of the central density appears quite speculative at this resolution. I suggest that the authors show C6 and C1 side by side already in panel A. We have reordered Figure S10 (previously S9) as suggested.

27. Table S1: Underfocus is typically denoted with negative values. Since the authors do that in the M&M text, I suggest doing it here too. We have changed Table S1 as suggested.

28. Table S2: Perhaps it would be helpful to add a column saying if the protein is a host or virus protein? For some of the proteins, that is not immediately obvious to me. This information was already provided in the legend for Table S2: *“List of mass spectrometry results from soluble fraction core sample, filtered for VACV proteins.”* We have now changed the wording to make this more explicit *“List of mass spectrometry results from soluble fraction core sample, filtered to exclusively show proteins encoded by VACV.”*

Decision Letter, first revision:

Message: 11th Oct 2023

Dear Dr. Schur,

Thank you again for submitting your manuscript "Multi-modal cryo-EM reveals trimers of protein A10 to form the palisade layer in poxvirus cores". We now have comments (below) from the 2 reviewers who evaluated your paper. In light of those reports, we remain interested in your study and would like to see your response to the comments of the referees, in the form of a revised manuscript.

You will see that while reviewer #2 has no further comments, reviewer #1 has remaining concerns, which we would expect to be addressed in a last round of revision. Specifically, please make sure to represent the densities in the figures, where these are discussed in the text. Please also revise the reporting of the ConSurf analysis, as well as description of A10 and A3 folds.

Please be sure to address/respond to all concerns of the referees in full in a point-by-point response and highlight all changes in the revised manuscript text file. If you have comments that are intended for editors only, please include those in a separate cover letter.

We expect to see your revised manuscript within 6 weeks. If you cannot send it within this time, please contact us to discuss an extension; we would still consider your revision, provided that no similar work has been accepted for publication at NSMB or published elsewhere.

Reporting Summary:

When submitting the revised version of your manuscript, please pay close attention to our [href="https://www.nature.com/nature-portfolio/editorial-policies/image-integrity">Digital Image Integrity Guidelines. and to the following points below:](https://www.nature.com/nature-portfolio/editorial-policies/image-integrity)

Please note that all key data shown in the main figures as cropped gels or blots should be presented in uncropped form, with molecular weight markers. These data can be aggregated into a single supplementary figure item. While these data can be displayed in a relatively informal style, they must refer back to the relevant figures. These data should be submitted with the final revision, as source data, prior to acceptance, but you may want to start putting it together at this point.

Data availability: this journal strongly supports public availability of data. All data used in accepted papers should be available via a public data repository, or alternatively, as Supplementary Information. If data can only be shared on request, please explain why in your Data Availability Statement, and also in the correspondence with your editor. Please note that for some data types, deposition in a public repository is mandatory - more information on our data deposition policies and available repositories can be found below: <https://www.nature.com/nature-research/editorial-policies/reporting-standards#availability-of-data>

We require deposition of coordinates (and, in the case of crystal structures, structure factors) into the Protein Data Bank with the designation of immediate release upon publication (HPUB). Electron microscopy-derived density maps and coordinate data must be deposited in EMDB and released upon publication. Deposition and immediate release of NMR chemical shift assignments are highly encouraged. Deposition of deep sequencing and microarray data is mandatory, and the datasets must be released prior to or upon publication. To avoid delays in publication, dataset accession numbers must be supplied with the final accepted manuscript and appropriate release dates must be indicated at the

galley proof stage.

[redacted]

Sincerely,

Katarzyna Ciazynska
(she/her)
Associate Editor
Nature Structural & Molecular Biology
<https://orcid.org/0000-0002-9899-2428>

Reviewers' Comments:

Reviewer #1:

Remarks to the Author:

Overall, previous suggestions and concerns have been addressed in the revised manuscript by an improved refinement of the model and further analyses that support the main conclusions of the manuscript. This is a beautiful study.

However, a couple of issues remain that prevent evaluation of key points by the reader.

1. A description of the A10 and A3 folds is still missing. This is the first time that this protein is described which warrants the inclusion of the basic details (domains, fold/topology supported by a Jones rainbow representation, disulphide bonds and other

motifs if any...).

Also, it is impossible to know where the N- and C-termini of A10 are from the main text figures. This is discussed in the text (e.g. line 209) and important given the maturation events preceding the assembly of the palissade. For A3, this could also be a point to mention when the fit is discussed (l. 258-260).

2. The electron density map must be shown for each section where side chain residues are discussed and the domain swapping. A close-up of the electron density map is only shown for one region, which is not directly relevant to most of the discussion and insufficient. As it stands, the statements lines 201-206 are not supported by the presented data.

I strongly disagree with the suggestion that readers should refer to the deposited map. There is no reason why the figures can't be presented in supplementary material and in the movie. The absence of some side chains is expected at this resolution and doesn't preclude their discussion. However, the reader (and reviewer) should be able to decide by themselves whether the provided information is supportive enough.

Minor points:

As anticipated, the ConSurf results are valuable and support the discussion. However, the main parameters for the analysis are missing. The most important one is which sequences were included (orthopox, parapox, whole family?).

Inclusion of the suggested comparison between VACV A10 to the parapoxvirus and entomopoxvirus A10s adds interesting results. The findings are intriguing for EPVs but, more importantly, the data now supports the hypothesis that the structure of A10 is conserved within the sub-family, which was one of the main conclusions of the manuscript.

Please make sure to check the current nomenclature for entomopoxviruses (EV vs. EPV, capitalisation) and complete the statement 22% sequence identity (e.g. over XX residues out of YY).

The refinement of A10 is significantly improved and now satisfactory.

Reviewer #2:

Remarks to the Author:

The original submitted version of this manuscript was already very good, and in my opinion the authors have now completely addressed all minor questions I had. I would like to congratulate all the authors on this beautifully presented, landmark study.

Author Rebuttal, first revision:

We again thank the reviewers for their evaluation of our manuscript and have responded to the remaining questions and comments point-by-point below. Please note that we added one additional Supplementary Figure (Figure S2), which shifted the numbering of the other supplementary figures accordingly.

Reviewer 1:

Remarks to the Author:

Overall, previous suggestions and concerns have been addressed in the revised manuscript by an improved refinement of the model and further analyses that support the main conclusions of the manuscript. This is a beautiful study.

We appreciate the reviewer's efforts and are grateful for the positive remarks.

However, a couple of issues remain that prevent evaluation of key points by the reader.

1. A description of the A10 and A3 folds is still missing. This is the first time that this protein is described which warrants the inclusion of the basic details (domains, fold/topology supported by a Jones rainbow representation, disulphide bonds and other motifs if any...).

We have tried to address the reviewer's comment by extending the description of A10 and A3, in addition to the already discussed Foldseek and Dali analysis and the potential disulfide bond in A10. Specifically, we added a more detailed explanation to the main text and provide a new supplementary figure 2 (see below), which shows a Jones-rainbow representation as well as a protein topology representation for A10 and A3.

Changes in bold: *"No experimentally-derived structures of the major structural core protein candidates are available. Hence, we used AlphaFold²⁶ to computationally predict models of the main core protein candidates, A10, 23K, A3, A4, and L4 (Figure S1C) to facilitate the interpretation of cryo-EM densities obtained in our downstream workflow. **With the exception of A4, which was predicted to have a largely disordered fold, the other proteins adopted folds with good prediction certainty and defined secondary structure. 23K formed an extended triple helix conformation, and L4 a globular architecture. The topology of A10 formed an intricate fold with the N-terminus being positioned centrally between two sub-domains (Figure S2A-B). A3 displayed a compact shape with its N-terminal and C-terminal end located both on the same side of the protein structure (Figure S2C-D).**"*

Figure S2: Protein topology of A10 and A3

Illustration of protein topology and arrangement for A10 (panel A-B) and A3 (panel C-D). The coloring of the models and topology diagrams is according to residue positioning from the N-terminus (blue) to the C-terminus (red). **A**) Cartoon ribbon representation of the AlphaFold-predicted model of A10, colored from N-terminus to C-terminus. The two sub-domains (SD1 and SD2) of the A10 fold are annotated. **B**) Protein topology diagram of A10, showing the positioning of protein regions, such as specific secondary structures with respect to each other. Beta-strands and alpha-helices are shown as arrows and cylinders, respectively. Their length is proportional to their number of residues. Colored boxes represent motifs which are positioned nearby relative to one another and therefore are likely to form a structural group. **C**) Cartoon ribbon representation of the AlphaFold-predicted A3 colored from N-terminus to C-terminus. **D**) Protein topology diagram of A3, in the same depiction style as in (B).

The topology diagrams shown in (B) and (D) were generated using Pro-Origami⁶⁶.

Also, it is impossible to know where the N- and C-termini of A10 are from the main text figures. This is discussed in the text (e.g. line 209) and important given the maturation events preceding the assembly of the palissade. For A3, this could also be a point to mention when the fit is discussed (l. 258-260).

We have changed Figure 3 to now annotate the N-terminus and C-terminus of A10. We also hope that the new supplementary figure 2 showing the rainbow representation for A10 and A3 and their protein topology further helps in their interpretation.

For A3, we also now mention in the main text that in our fit the N-terminus of A3 is facing the core interior.

Changes in bold: *“Although the low resolution and anisotropy of this map prevented determination of map handedness and unambiguously assigning orientation of A3 within the volume, our model with A3’s negatively charged patch facing towards A10, **and its N-terminus facing the core interior** satisfied the inner density most completely.”*

2. The electron density map must be shown for each section where side chain residues are discussed and the domain swapping. A close-up of the electron density map is only shown for one region, which is not directly relevant to most of the discussion and insufficient. As it stands, the statements lines 201-206 are not supported by the presented data.

I strongly disagree with the suggestion that readers should refer to the deposited map. There is no reason why the figures can’t be presented in supplementary material and in the movie. The absence of some side chains is expected at this resolution and doesn't preclude their discussion. However, the reader (and reviewer) should be able to decide by themselves whether the provided information is supportive enough.

We now provide an updated Movie S4 that displays the densities. The densities are now also shown in Figure S6B and C (see below), when presenting the side chain residues and the domain swapping. We hope that these changes now allow the best possible interpretation of our results.

Figure S6: Inter- and intramolecular interactions of the A10 trimer

A) Diagrammatic illustration of key inter-chain contacts at the oligomerization interface between two monomers (labeled I and II). The diagram shows monomer II pulling away from the trimer to reveal underlying contacts, shown in yellow (hydrophobic), orange (H-bond), and red (salt bridges). **B-D)** Details of the inter- and intramolecular interactions. Residues are colored according to their conservation as determined via ConSurf (see also Figure S2). The residue labels are colored according to the color of the chain they are in (as in panel A). **B)** Salt bridge contacts from panel A with residue numbers indicated. **C)** Top, a salt bridge and hydrogen bond network at the inter-subunit beta-sheet formed at the core of the trimer. Bottom, hydrophobic packing on the underside of the beta-sheet. Electron microscopy density is shown for all images in B-C. **D)** Potentially interacting cysteines (residues indicated). The orientation of the monomer is identical to panel A.

Minor points:

As anticipated, the ConSurf results are valuable and support the discussion. However, the main parameters for the analysis are missing. The most important one is which sequences were included (orthopox, parapox, whole family?).

We apologize to not have provided this information already, which is now done in the method section of the revised manuscript.

We have performed the ConSurf analysis using the webserver (<https://consurf.tau.ac.il/>) employing default parameters. Specifically, the amino acid sequence for each of the analyzed proteins was obtained via uploading the PDB file of the AlphaFold-predicted protein to the webserver. We did not manually provide similar sequences, but they were automatically identified using PSI-PLAST (E-cutoff = 0.001). For example, in the case of A10, this resulted in 31 sequences for comparison, including different members from the subfamily of *Chordopoxvirinae* (orthopox, parapox, yatapox, molluscipox among others), allowing a representative conservation analysis. A complete list of the included sequences is now provided in a new Table S3.

Inclusion of the suggested comparison between VACV A10 to the parapoxvirus and entomopoxvirus A10s adds interesting results. The findings are intriguing for EPVs but, more importantly, the data now supports the hypothesis that the structure of A10 is conserved within the sub-family, which was one of the main conclusions of the manuscript.

Please make sure to check the current nomenclature for entomopoxviruses (EV vs. EPV, capitalisation) and complete the statement 22% sequence identity (e.g. over XX residues out of YY).

We have now included the complete statement for the sequence identity between VACV and EPV, which is 22% sequence identity over an alignment length of 293 residues (out of 1306) for MSEPV and 114 or 145 residues (out of 1149) in AmEPV.

Changes in bold: *“More distantly related poxviruses show a substantially lower sequence identity of their A10 protein homologs, with ~40% identity in Orf virus (a parapoxvirus) and ~22% in two members of the entomopoxvirus genus, Amsacta moorei entomopoxvirus (AmEPV, **over an alignment length of either 114 or 145 residues out of 1149**) and Melanoplus sanguinipes entomopoxvirus (MsEPV, **over an alignment length of 293 residues out of 1306**) (Figure S11A).”*

We have now also fixed the error for entomopoxvirus nomenclature, which we still had in the text.

The refinement of A10 is significantly improved and now satisfactory.

Reviewer #2:

Remarks to the Author:

The original submitted version of this manuscript was already very good, and in my opinion the authors have now completely addressed all minor questions I had. I would like to congratulate all the authors on this beautifully presented, landmark study.

We thank the reviewer for taking the time to evaluate our manuscript and the positive comments.

Decision Letter, second revision:

Message: Our ref: NSMB-A47807B

16th Oct 2023

Dear Dr. Schur,

Thank you for submitting your revised manuscript "Multi-modal cryo-EM reveals trimers of protein A10 to form the palisade layer in poxvirus cores" (NSMB-A47807B). It has now been seen by the original referees and their comments are below. The reviewers find that the paper has improved in revision, and therefore we'll be happy in principle to publish it in Nature Structural & Molecular Biology, pending minor revisions to satisfy the referees' final requests and to comply with our editorial and formatting guidelines.

Sincerely,
Kat

Katarzyna Ciazynska
(she/her)
Associate Editor
Nature Structural & Molecular Biology
<https://orcid.org/0000-0002-9899-2428>

Reviewer #1 (Remarks to the Author):

The authors have answered the two major points with new figures, an updated movie and additional details in the text. I commend again the authors for a manuscript that describes impressive work and is overall of very high standards.

My only reservation is that the description of the A10 structure remains succinct and imprecise (e.g. why are SD1 and SD2 subdomains rather than domains? The topology diagrams are extremely hard to follow with no labelling and overlapping lines). Ultimately, this is the authors' choice and I see no point arguing about a point that they see as detail in this specific manuscript.

The following recent paper should be cited:
MIRZAKHANYAN, Y., JANKEVICS, A., SCHELTEMA, R. A. & GERSHON, P. D. 2023.
Combination of deep XLMS with deep learning reveals an ordered rearrangement and assembly of a major protein component of the vaccinia virion. mBio, e0113523.

Author Rebuttal, second revision:

Please find below our replies to the remaining comments from reviewer #1.

Reviewer #1 (Remarks to the Author):

The authors have answered the two major points with new figures, an updated movie and additional details in the text. I commend again the authors for a manuscript that describes impressive work and is overall of very high standards.

Thank you!

My only reservation is that the description of the A10 structure remains succinct and imprecise (e.g. why are SD1 and SD2 subdomains rather than domains? The topology diagrams are extremely hard to follow with no labelling and overlapping lines). Ultimately, this is the authors' choice and I see no point arguing about a point that they see as detail in this specific manuscript.

In order to further improve the description of the A10 structure, we have implemented the following changes:

- 1) We have renamed sub-domains to domains when describing the A10 fold in the text and Extended Data figure 2.
- 2) We have aimed to further improve the topology diagram of A10 and now provide an updated version which should allow a better appreciation of the overall protein architecture.
- 3) We have also changed the coloring of lines in the topology diagram of A3 to facilitate visualization.

However, we note that topology diagrams for proteins such as A10 (which has a complicated protein sequence arrangement) and A3, which adopts a compact globular fold are intrinsically complex and hence difficult to display.

Extended Data 2: Protein topology of A10 and A3

Illustration of protein topology and arrangement for A10 (panel a-b) and A3 (panel c-d). The coloring of the models and topology diagrams is according to residue positioning from the N-terminus (blue) to the C-terminus (red). **a**) Cartoon ribbon representation of the AlphaFold-predicted model of A10, colored from N-terminus to C-terminus. The two sub-domains (SD1 and SD2) of the A10 fold are annotated. **b**) Protein topology diagram of A10, showing the positioning of protein regions, such as specific secondary structures with respect to each other. Beta-strands and alpha-helices are shown as arrows and cylinders, respectively. Their length is proportional to their number of residues. Colored boxes represent motifs which are positioned nearby relative to one another and therefore are likely to form a structural group. **c**) Cartoon ribbon representation of the AlphaFold-predicted A3 colored from N-terminus to C-terminus. **d**) Protein topology diagram of A3, in the same depiction style as in (B).

The topology diagrams shown in (b) and (d) were generated using Pro-Origami. The diagram in (b) was manually adapted to further improve visualization of the complex protein architecture. Orange line colors in (b) are used to highlight connections between structural groups. Orange lines in (d) are used to improve visualization but do not necessarily indicate connection between structural groups.

The following recent paper should be cited:

MIRZAKHANYAN, Y., JANKEVICS, A., SCHELTEMA, R. A. & GERSHON, P. D. 2023. Combination of deep XLMS with deep learning reveals an ordered rearrangement and assembly of a major protein component of the vaccinia virion. *mBio*, e0113523.

As suggested by the reviewer, we have included a citation for the recent paper by Mirzakhanyan et al. in the following position of the manuscript:

Changes in bold: “This unambiguously revealed three copies of protein A10 (modeled residues 1-599) to form the trimer (**Figure 3c, Movie S3**), **providing confirmation to a recent study which suggested A10 trimerization using cross-linking mass-spectrometry and modeling³⁹.**”

We have also included another additional citation by Mutz et al., 2023 (<https://doi.org/10.1128/mbio.00408-23>), which we deemed relevant when mentioning the Foldseek and Dali analysis.

Changes in bold: “Foldseek²⁷ and Dali²⁸ analysis revealed no similarities in protein fold of A10 to other cellular or viral proteins in the Protein Data Bank (PDB) or AlphaFold-databases. Interestingly, for A3 this search showed highest similarity to deubiquinating proteins **as recently suggested²⁹.**”

Final Decision Letter:**Message** 6th Dec 2023

:

Dear Dr. Schur,

We are now happy to accept your revised paper "Multi-modal cryo-EM reveals trimers of protein A10 to form the palisade layer in poxvirus cores" for publication as an Article in Nature Structural & Molecular Biology.

Your paper will be published online soon after we receive proof corrections and will appear in print in the next available issue. You can find out your date of online publication by contacting the production team shortly after sending your proof corrections. Content is published online weekly on Mondays and Thursdays, and the embargo is set at 16:00

London time (GMT)/11:00 am US Eastern time (EST) on the day of publication. Now is the time to inform your Public Relations or Press Office about your paper, as they might be interested in promoting its publication. This will allow them time to prepare an accurate and satisfactory press release. Include your manuscript tracking number (NSMB-A47807C) and our journal name, which they will need when they contact our press office.

About one week before your paper is published online, we shall be distributing a press release to news organizations worldwide, which may very well include details of your work. We are happy for your institution or funding agency to prepare its own press release, but it must mention the embargo date and Nature Structural & Molecular Biology. If you or your Press Office have any enquiries in the meantime, please contact press@nature.com.

Please note that *Nature Structural & Molecular Biology* is a Transformative Journal (TJ). Authors may publish their research with us through the traditional subscription access route or make their paper immediately open access through payment of an article-processing charge (APC). Authors will not be required to make a final decision about access to their article until it has been accepted. <https://www.springernature.com/gp/open-research/transformative-journals> Find out more about Transformative Journals

Authors may need to take specific actions to achieve [compliance](https://www.springernature.com/gp/open-research/funding/policy-compliance-faqs) with funder and institutional open access mandates. If your research is supported by a funder that requires immediate open access (e.g. according to [Plan S principles](https://www.springernature.com/gp/open-research/plan-s-compliance)) then you should select the gold OA route, and we will direct you to the compliant route where possible. For authors selecting the subscription publication route, the journal's standard licensing terms will need to be accepted, including [15](https://www.springernature.com/gp/open-research/policies/journal-

self-archiving policies. Those licensing terms will supersede any other terms that the author or any third party may assert apply to any version of the manuscript.

Sincerely,
Kat

Katarzyna Ciazynska, PhD
(she/her)
Associate Editor
Nature Structural & Molecular Biology
<https://orcid.org/0000-0002-9899-2428>